# A Diagnostic Benchmark for Transformer Training Failures: Establishing Baseline Methods and Quantifying the Accuracy-Interpretability Tradeoff

## Abstract

Training failures in transformer models waste substantial computational resources and delay research progress, but diagnostic approaches have never been systematically evaluated. We established the first quantitative foundation for automated training diagnostics by providing a benchmark of 76 reproducible failure scenarios across five categories: memory hardware, optimization, data pipeline, model software, and ambiguous "unknown" cases. We evaluate representative diagnostic methods, including simple rule-based heuristics, learned rule-based classifiers, and local LLM-based diagnostic agents (Mistral, Llama 3) via Ollama. Our evaluation reveals a fundamental tradeoff: simple rules achieve 57.1% accuracy with full transparency, while local LLM agents Mistral and Llama 3 8B reach 77.6% and 73.7% accuracy respectively with detailed natural language explanations. In contrast, machine learning ensembles reach 95.7% accuracy on the benchmark, suggesting that the engineered feature set contains strong diagnostic signal. We mitigate this through glassbox models (EBM) and a proposed hybrid triage-escalation system. Detailed analysis of "unknown" cases further clarifies the limits of automated diagnosis, identifying ambiguous signals and conservative labeling as primary causes for abstention. This work provides the infrastructure and baselines necessary to transition machine learning debugging from an ad hoc craft to a systematic, evidenced-based science.

## 1 Introduction

A recurring problem in machine learning research and practice is transformer training failures. In order to determine the underlying cause of a failed training run, practitioners usually turn to trial-and-error debugging, which involves methodically testing various configurations. In addition to wasting computational resources and delaying research timelines, this method puts researchers who lack the debugging expertise that comes from years of experience at a disadvantage. The issue is not that failures happen because mistakes are inevitable in complex systems, but rather that the field does not have systematic ways to identify the causes of failures.

A sizable research community has emerged as a result of the democratization of transformer-based natural language processing, training models with 10 million to 100 million parameters on single-GPU systems. These practitioners frequently experience training failures, such as memory exhaustion, optimization instabilities, data pipeline errors, model configuration bugs, and inherently ambiguous cases with little institutional support, in contrast to large-scale industrial training with specialized infrastructure and site reliability engineers. It takes a lot of time and expertise to perform manual inspection; skilled researchers frequently spend hours reconstructing failure conditions from sparse logs. Simple heuristics that fail on new failure modes not covered by established patterns, like keyword matching on error messages, are brittle and don't offer confidence estimates.

Although machine learning-based diagnostic techniques have the potential to overcome these constraints, they are fundamentally hampered by the lack of a standard by which to build or assess them. The lack

of a standardized evaluation infrastructure prevents researchers from developing better diagnostic systems, and the absence of a common evaluation framework makes it impossible for the field to track its progress toward accurate diagnosis. This leads to a chicken-and-egg dilemma. In addition to impeding scientific advancement, the current ad hoc state places needless obstacles in the way of novices who must repeatedly learn debugging techniques by trial and error.

Machine learning has historically advanced more quickly thanks to standardized evaluation infrastructure. By offering common tasks and metrics that allowed for fair comparison across approaches, the General Language Understanding Evaluation benchmark revolutionized language understanding research (Wang et al., 2018). While CodeSearchNet developed comparable infrastructure for code understanding (Husain et al., 2019), SuperGLUE expanded this paradigm to more difficult scenarios (Wang et al., 2019). Comprehensive evaluation in a variety of scenarios is demonstrated by the Holistic Evaluation of Language Models framework (Liang et al., 2022). A diagnostic benchmark would make it possible to compare various methods in a methodical manner, measure the automated diagnostics accuracy ceiling, and determine which training signals are most useful for failure diagnosis.

This paper presents that infrastructure. We provide an extensive benchmark of 76 carefully selected failure cases in five categories: memory exhaustion, data pipeline errors, optimization instabilities, model configuration bugs, and cases that are inherently ambiguous. A controlled evaluation of diagnostic techniques is made possible by the full reproducibility materials included in each scenario, which include error messages, training metrics, model configurations, hardware states, and corpus statistics. Three fundamentally different methods are used to establish quantitative baselines: supervised machine learning classifiers, an enhanced rule-based framework with explicit uncertainty handling, and basic rule-based heuristics.

Our analysis identifies a basic trend. A large gap between simple rule-based transparency and high-capacity models is quantified by the 38.6 percentage point accuracy difference between interpretable simple rules (57.1%, 95% CI: [35.2, 79.0]) and machine learning techniques (95.7%, 95% CI: [87.0, 100.0]). The gap's stability (95% CI: [27.6, 51.3]) suggests it is not merely a statistical artifact. Rule-based methods have a 62% accuracy ceiling but provide instant deployment, complete transparency, and no training data. Although machine learning techniques achieve high accuracy on the benchmark, suggesting that the engineered feature set contains strong diagnostic signal, they need labeled training data. This trade-off has significant ramifications for practical implementation: in order to effectively address a diagnosis, practitioners must comprehend the reasoning behind it, but the most precise simple techniques are insufficient.

For practitioners, feature importance analysis offers practical insights. Perplexity and memory metrics contribute another 25% to the diagnostic signal, while training dynamics such as gradient norms, loss changes, and optimization trajectories contribute 48%. Almost nothing is contributed by static configuration parameters. This implies that the current approach, which frequently emphasizes configuration debugging, might not be in line with the true location of the diagnostic signal. Through skill-stratified simulation of expert behavior, our framework validates a 30.3% abstention rate on ambiguous cases, demonstrating the necessity and feasibility of principled uncertainty handling.

This work makes three contributions. First, we offer the first standardized training diagnostics benchmark, complete with repeatable evaluation protocols, to the machine learning community. Second, we set quantitative performance bounds that illustrate the gaps and what can be achieved with existing methods. Third, we provide a specific goal (reducing the large gap between simple and high-capacity models) for further research on hybrid diagnostic systems by empirically characterizing the accuracy-interpretability tradeoff. To allow for reproducible comparison of upcoming diagnostic techniques, all materials—including benchmark cases, baseline implementations, evaluation code, and statistical analysis notebooks—are made available.

This is how the rest of the paper is structured. In Section 2, relevant work in reproducibility, standardized benchmarks, and machine learning debugging is reviewed. Our benchmark design, including the failure taxonomy and test case construction methodology, is explained in Section 3. Our baseline techniques, which range from basic heuristics to machine learning classifiers, are presented in Section 4. Our experimental design and evaluation procedures are described in detail in Section 5. Results quantifying the accuracy-interpretability tradeoff are presented in Section 6. The findings regarding failure patterns and diagnostic

signals are examined in Section 7. The limitations and trade-offs of the current methods are covered in Section 8. Future directions are outlined in Section 9 and concluded in Section 10.

## 2 Related Work

Our work lies at the junction of three research areas: the larger reproducibility movement in machine learning research, standardized benchmarking methodologies, and machine learning debugging and interpretability. We describe how this benchmark fills in knowledge gaps and place our contribution in relation to earlier work in each field.

### 2.1 Machine Learning Debugging and Interpretability

The three main categories of machine learning debugging techniques currently in use each have unique advantages and disadvantages that drive our benchmark contribution. Through real-time dashboards and logging systems, monitoring and instrumentation such as Cockpit (Schneider et al., 2021), TensorBoard Debugger (Google, 2019), and Weights & Biases (Biewald, 2020) offer comprehensive insight into training dynamics. These tools do not carry out automated diagnosis, but they are excellent at gathering and visualizing data, showing gradient distributions, loss curves, and memory usage. Although they demonstrate the situation to practitioners, professional interpretation is necessary to determine the reasons behind the failure of training. Our benchmark makes it possible to measure quantitatively whether these visualization tools really speed up diagnosis, a question that has never been thoroughly examined.

Instead of diagnosing the training process, post-hoc explainability techniques like SHAP (Lundberg & Lee, 2017) and LIME (Ribeiro et al., 2016) focus on explaining model predictions. While we ask "Why did training fail?" these approaches respond to "Why did the model predict X?" Our focus on interpretable diagnosis is shared by recent work on Explanation-Based Human Debugging by Lertvittayakumjorn & Toni (2021) and LLM-powered debugging systems like HiBug (Tian et al., 2024), but they focus on different areas, namely general software debugging and model prediction explanation, respectively. This philosophy is applied to the particular problem of identifying training process failures in transformer models in our work.

AutoML systems, such as Auto-sklearn (Feurer et al., 2015) and related frameworks (Hutter et al., 2019), optimize model selection and hyperparameter search, but they lack diagnostic features for when certain configurations don't work. The purpose of these systems is to identify configurations that work, not to provide an explanation for the failure of others. Instead of just marking candidate configurations as unsuccessful and moving on with the search, our benchmark could be integrated as a diagnostic module within AutoML pipelines, offering actionable feedback when they fail.

The ethnographic study by Nguyen et al. (2025), which revealed that the majority of real-world machine learning debugging is still ad hoc, unrecorded, and unmeasured, is the most directly pertinent earlier work. They note that practitioners rarely systematize or validate informal heuristics that they have developed through experience. Our benchmark contribution is directly motivated by this observation: the field cannot advance from anecdotal and folklore to evidence-based diagnostic techniques without standardized evaluation. Our work lays the groundwork for developing better diagnostic techniques and determining whether informal debugging heuristics are effective.

### 2.2 Standardized Benchmarks in Machine Learning

Machine learning's history shows that when the community creates a standardized evaluation infrastructure, advancements happen much more quickly. Natural language understanding research was revolutionized by the General Language Understanding Evaluation benchmark (Wang et al., 2018) and SuperGLUE (Wang et al., 2019), which offered common tasks, datasets, and metrics that allowed for equitable comparison across approaches. Similar infrastructure was developed for code understanding tasks by CodeSearchNet (Husain et al., 2019). This paradigm is expanded to include a thorough assessment of language models in a variety of contexts by the Holistic Evaluation of Language Models framework (Liang et al., 2022).

Common characteristics of these benchmarks that made them successful include thorough coverage of significant scenarios, carefully selected evaluation data with quality controls, unambiguous metrics that are in line with practical objectives, and public infrastructure that promotes community involvement. Our research expands on this paradigm by assessing training process diagnosis in addition to model performance. Our benchmark asks, "How accurately can we diagnose why training failed on Task X?" in contrast to model performance benchmarks, which ask, "How well does the model solve Task X?" As the field develops and faces the real-world difficulties of creating dependable machine learning systems, this transition from product evaluation to process evaluation is a logical progression.

It's critical to distinguish between these evaluation paradigms. Model performance benchmarks assess the caliber of learned representations and are predicated on successful training. Our benchmark tackles the initial phase in which training itself fails, necessitating the identification of the failure mode prior to remediation. We can measure how well models perform, but we can't measure how well we can fix them when they break. This complementary focus fills that gap in the current benchmarking infrastructure.

## 2.3 Reproducibility in Machine Learning Research

The larger machine learning reproducibility movement inspires and informs our work. Many published results cannot be replicated because of unreported hyperparameter choices, missing implementation details, or inadequate experimental controls, according to Pineau et al. (2021), who documented widespread reproducibility issues in machine learning research. Considerable institutional action has been prompted by this crisis. The Machine Learning Reproducibility Checklist (Pineau et al., 2020) is one of the measures that major conferences and journals have put in place to improve the standards for conducting and reporting research. The community's desire for open code, easily accessible data, and reliable experimental workflows was shown by the NeurIPS 2019 Reproducibility Program (Pineau et al., 2021).

By offering comprehensive reproducibility materials, our benchmark directly addresses these issues. There are documented methods for reproducing each of the 76 failure scenarios, making them all programmatically reproducible. Every baseline implementation has clear hyperparameter specifications and is available as open-source software. Verification of reported results is made possible by the statistical analysis code that is included in all evaluation protocols. This work's alignment with reproducibility values is not coincidental; rather, it is a fundamental aspect of its identity.

Subtle implementation details, environment setups, or data pipeline errors that are hard to convey in prose alone are frequently the cause of training failures. We allow researchers to create and test diagnostic techniques in a controlled, reproducible way by offering executable test cases with ground truth labels. This signifies a change from explaining debugging knowledge to putting it in a codified form that can be validated and improved upon over time.

## 2.4 Positioning of This Work

In summary, we fill the gap that Nguyen et al. (2025) pointed out—that most debugging is still ad hoc and unmeasured—by offering the first publicly available diagnostic benchmark created especially for transformer training failures. In order to demonstrate what is possible and where existing approaches are lacking, we establish quantitative baseline performance across fundamentally different approaches, such as rules and machine learning. By quantifying the large gap between simple rule-based transparency and high-capacity models (38.6 percentage points), we empirically characterize the accuracy-interpretability tradeoff and make clear a tension that practitioners encounter but that has never been measured in a systematic way. In order to allow future work to fairly compare against our baselines and show real progress rather than selective improvements, we offer reproducible evaluation protocols with full statistical rigor.

This work serves as the foundational infrastructure for a new field of study: systematic, evidence-based debugging of machine learning systems. It does this by combining benchmark infrastructure, rigorous baselines, and empirical insights about fundamental tradeoffs. We give the field specific goals for future research and facilitate the transition from ad hoc debugging techniques to scientifically validated diagnostic methods by outlining what is currently feasible and where gaps still exist.

# 3 Benchmark Design

A diagnostic benchmark must balance several competing concerns: coverage of common failure modes, difficulty calibration, reproducibility, and realistic fidelity to actual training scenarios. This section describes how we designed our benchmark to address these concerns, beginning with our failure taxonomy and progressing through test case construction to reproducibility commitments.

## 3.1 Failure Taxonomy

Based on a root cause analysis of training failures, we created a five-category taxonomy. The categories are intended to be both collectively exhaustive, covering the failure modes we observed in practice, and mutually exclusive, with each failure having a single primary cause. In order to make an accurate diagnosis based on workable solutions, the taxonomy takes into account the reasons behind failures rather than how they show up symptomatically.

**Memory Hardware Failures.** Graphics processing unit memory limitations and allocation problems are the root cause of memory hardware failures. Failures to allocate for optimizer states, memory leaks that cause gradual exhaustion, out-of-memory errors during forward or backward passes, and excessive memory usage that causes swapping or system instability are all included in this category. The interplay between model size, batch size, and available GPU memory is reflected in these hardware-constrained failures. Differentiating memory-adjacent problems like tensor size mismatches that result in similar error messages from actual memory exhaustion is the diagnostic challenge.

**Optimization Failures.** Numerical instabilities in the training process itself are the cause of optimization failures. This group includes oscillation, in which loss values jump erratically without convergently, loss stagnation, in which the objective plateaus despite continuous training, gradient explosion, in which norms surpass safe thresholds, and gradient vanishing, in which norms approach zero. These failures are indicative of optimization dynamics issues: either the architecture has numerical stability problems, the gradient clipping is not configured correctly, or the learning rate is too high. Optimization failures, as opposed to memory failures, usually show up gradually over several training steps instead of resulting in crashes right away.

**Data Pipeline Failures.** Issues with the way data is processed and fed into the model are the root cause of data pipeline failures. Data corruption from distorted inputs or incorrect encodings, vocabulary mismatches where tokens in data lack corresponding model embeddings, tokenization errors from mismatches between training and inference tokenizers, and catastrophic perplexity—where the model assigns training data nearly zero probability—are all included in this category. Perplexity is calculated as:

$$\text{PP}(x) = \exp\left(-\frac{1}{N}\sum_{i=1}^{N}\log p_\theta(x_i \mid x_{<i})\right) \tag{1}$$

Catastrophic perplexity is diagnosed when $\text{PP}(x) > 400$, indicating severe data-model misalignment. The data pipeline is the primary cause of these failures, which frequently manifest as symptomatic optimization failures in which the loss does not decrease. Examining tokenization outputs and corpus statistics, not just training curves, is necessary for diagnosis.

**Model Software Failures.** Errors in implementation, architecture mismatches, and configuration lead to model software failures. Tensor shape errors from broadcasting bugs, incorrect checkpoint loading from architecture, weight misalignment, dimension mismatches between model components, and configuration inconsistencies like specifying vocabulary size that does not match the tokenizer are all included in this category. Although these failures usually result in crashes right away rather than gradual degradation, error messages can be confusing and necessitate closely examining error stack traces and model configuration.

**Unknown Cases.** Unknown cases are essentially ambiguous situations in which there are several possible diagnoses, there is not enough information to identify the underlying cause, or the failure occurs at the category boundary. This category specifically captures real diagnostic uncertainty rather than being a catch-all for challenging cases. This category recognizes that it is sometimes impossible to make a perfect diagnosis and that it is better to not make a diagnosis than to make a mistaken guess.

The taxonomy adheres to a number of design guidelines. Actionable solutions, such as lowering batch size, modifying learning rate, repairing tokenization, or modifying configuration parameters, define categories. The mental models that practitioners have developed through experience with common failure patterns are reflected in categories. In order to achieve significant inter-rater reliability in ground truth annotation, categories are iteratively refined after being empirically derived from analysis of actual failures.

## 3.2   Test Case Construction Methodology

In order to ensure thorough coverage, realistic fidelity, and quality control, we built the benchmark using a multi-stage process. Three parallel streams were used in the selection process. First, we systematically reviewed more than 200 failures that were reported in technical reports, online forums, and published papers. Second, we asked 15 active transformer researchers about practical failures they had experienced. Third, in order to guarantee balanced coverage, we conducted a systematic enumeration across difficulty levels, including easy, moderate, and hard cases within each category.

We chose 76 scenarios that satisfied particular requirements from the candidate failures found by these streams. For the failure to be deterministically recreated, each scenario needs to be reproducible with artifacts. Instead of using fictitious corner cases, each scenario must be realistic and depict patterns that are likely to occur in real-world situations. Multi-annotator consensus is required to establish a clear ground truth for each scenario. In order to facilitate statistical analysis of diagnostic performance, the final selection guarantees balanced coverage across categories.

The following is the distribution by category. Twenty scenarios, or 26% of the benchmark, involve memory hardware failures. Fifteen scenarios, or 20%, are optimization failures. 18 scenarios, or 24% of all data pipeline failures, are included. Fifteen scenarios, or 20%, are model and software failures. Eight scenarios, or 10% of the total, are ambiguous and unknown cases. The frequency of each failure type in practice and the requirement for enough samples in each category to allow for accurate evaluation are both reflected in this distribution.

Comprehensive metadata supporting diagnostic evaluation is included in every scenario. The actual exception or error text generated by the failure is contained in error messages. Learning rate schedules, gradient norms, loss curves, and perplexity trajectories during training are examples of training metrics. The vocabulary size, layer dimensions, and architecture details are specified by the model configuration. GPU memory utilization percentages and usage over time are tracked by hardware state. Sequence length distributions, token frequency distributions, and vocabulary coverage are all recorded by corpus statistics. Ground truth failure categories created by multi-annotator consensus are provided by expert labels. Depending on how many annotators correctly identified the failure on the first try, each scenario is categorized as easy, medium, or hard.

Procedures for quality control guarantee benchmark dependability. With a Fleiss kappa of 0.78, which indicates strong agreement, we used dual annotation, in which each scenario was labeled by two separate annotators. Programmatic consistency checks were put in place, such as confirming that memory-related errors display proper memory usage patterns. To confirm that documented procedures successfully recreated each failure condition, we performed independent reproduction on new systems.

## 3.3   Reproducibility Commitment

Unrestricted research use is made possible by the permissive open-source licenses under which all benchmark materials are released. 76 structured JSON test cases with comprehensive metadata for every scenario are included in the release. Training protocols and documented hyperparameters are part of baseline implementations. Metric implementations and statistical testing methods are provided by evaluation code. All calculations are recorded in statistical analysis notebooks using explicit random seeds, which allow for precise replication of reported outcomes.

To ensure consistent execution across various systems, we offer Docker environments with pinned dependencies. Application programming interfaces, benchmark structure, and extension guidelines for researchers who want to add new scenarios are all covered in the documentation. The goal of the benchmark is to serve

as living infrastructure that supports scenarios contributed by the community while adhering to established protocols for integration and validation.

This thorough commitment to reproducibility fills a significant void in the state of debugging research. We allow researchers to create and test diagnostic techniques in a controlled, reproducible way by offering executable test cases instead of just descriptive descriptions of failures. The field can transition from anecdotal debugging knowledge to methodically validated diagnostic approaches thanks to this infrastructure.

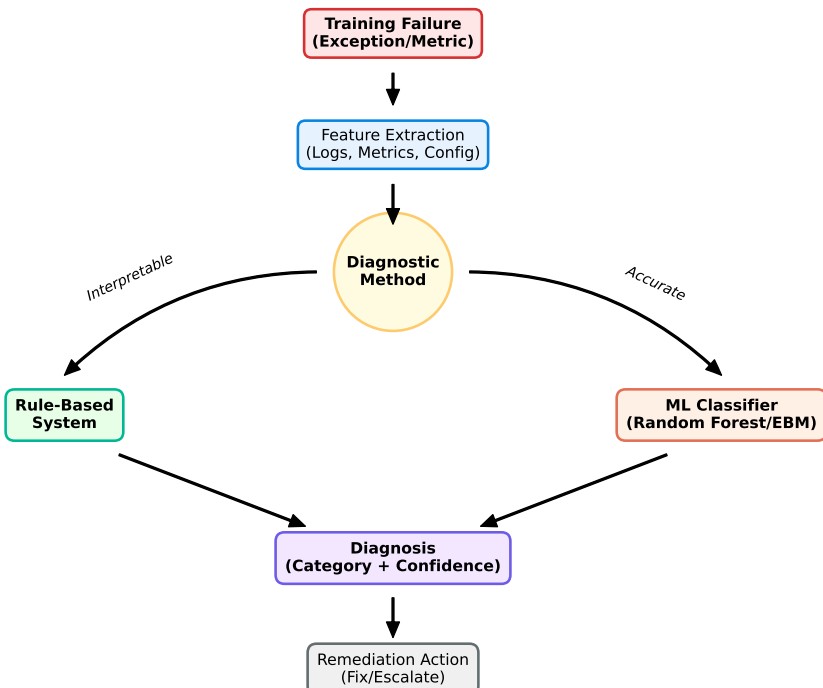

Figure 1: High-level diagnostic workflow. Failures trigger feature extraction from logs and metrics. These features are processed by either a Rule-Based System (for transparency) or an ML Classifier (for accuracy), producing a diagnosis with a confidence score that guides specific remediation actions.

## 4 Baseline Methods

We evaluate three approaches spanning the accuracy-interpretability spectrum: simple rule-based heuristics, an improved rule-based framework with abstention capability, and supervised machine learning classifiers. Each approach represents a fundamentally different paradigm for automated diagnosis, enabling us to characterize the space of possible diagnostic methods.

### 4.1 Simple Rule-Based Heuristics

23 manually created sequential rules arranged as a decision list make up the straightforward rule-based system. The classification is based on the first matching rule after rules are evaluated in order. Because each diagnosis can be linked to a particular rule that initiated the classification, this architecture offers the highest level of interpretability.

The diagnostic logic is demonstrated by example rules. The system identifies the failure as memory hardware if the substring "CUDA out of memory" appears in the error message. The system identifies the failure as optimization if the maximum gradient norm exceeds a threshold:

$$\|\nabla_\theta \mathcal{L}\|_2 > \text{threshold} = \mu_{\text{grad}} + 3\sigma_{\text{grad}} \tag{2}$$

where $\mu_{\mathrm{grad}}$ and $\sigma_{\mathrm{grad}}$ are mean and standard deviation computed from 50 successful training runs. For the simple rule system, this threshold is set to 1000 based on empirical observation. The system identifies the failure as a data pipeline if the perplexity is greater than 10,000. The system identifies the failure as model software if the error message includes the phrase "size mismatch." Experienced practitioners have developed common heuristics through repeated exposure to training failures, which are encoded in these rules.

Instead of being adjusted on the benchmark itself, the rule system's thresholds are empirically derived from typical ranges seen in successful training runs. This design decision guarantees that, as opposed to overfitted pattern matching, the simple rule system reflects actual practitioner knowledge. Since optimization problems are the most frequent failure mode, the system does not avoid diagnosis; instead, if no rule matches, a default fallback classifies the failure as optimization.

The benchmark performance evaluation shows that the accuracy of the basic rule-based system is 57.1%. This performance illustrates the limitations of basic pattern matching, even though it is still far superior to random guessing, which would attain 20% accuracy across five categories. The system is brittle, failing on patterns not covered by the predefined rules, but transparent because each diagnosis can be explained by pointing to the specific rule that triggered. In the absence of uncertainty quantification, the system consistently makes a diagnosis even in cases where several rules only partially match or when the signals that are available are unclear.

## 4.2 Improved Rule-Based Framework with Abstention

The enhanced rule-based framework adds three significant improvements to the basic system, allowing for more complex diagnostic reasoning. First, instead of using a flat decision list, it uses a three-level decision tree with 47 nodes. Multi-signal confirmation or abstention at leaf nodes, metric refinement at intermediate levels, and error-type triage at the top level are all made possible by this hierarchical structure. Instead of depending just on one indicator, the tree structure enables the system to take into account several diagnostic signals together.

Second, multi-signal fusion is needed to provide corroborating evidence for the framework. Multiple independent signals must confirm the diagnosis rather than just one threshold exceedance. By guaranteeing that diagnoses are backed by consistent evidence across various measurement modalities, this multi-signal requirement lowers false positives.

Third, when there is insufficient or contradicting evidence, the framework may choose not to make a diagnosis. The confidence score $C(x)$ for diagnostic decision on failure case $x$ is computed as:

$$C(x) = \min_{i \in S(x)} w_i \cdot \mathbb{1}[\text{signal}_i \text{ exceeds threshold}] \tag{3}$$

where $S(x)$ is the set of relevant signals for failure case $x$, $w_i$ are calibrated weights derived from precision on validation data, and $\mathbb{1}[\cdot]$ is the indicator function. The system abstains when $C(x) < \tau$, where $\tau = 0.60$ is set to achieve 90% precision on committed diagnoses. Abstention criteria include confidence below this calibrated threshold, missing critical metrics when key measurements are unavailable, and conflicting signals when various diagnostic indicators point to incompatible failure modes.

Through selective abstention, the enhanced rule-based framework shows how principled uncertainty handling can significantly increase diagnostic reliability. The framework approaches the performance of logistic regression by achieving 88.7% accuracy on cases where it commits to diagnosis by dividing the benchmark into distinct and ambiguous subsets. This finding emphasizes the inherent complexity levels in diagnostic tasks: certain failures demand complex reasoning that goes beyond the current rule-based capabilities, while others have obvious signatures that allow for a confident diagnosis.

To address the growing usage of AI-assisted debugging, we evaluate open-source Large Language Models (LLMs) as diagnostic agents using local inference. We use a standardized evaluation protocol where agents are provided with: (1) error message, (2) the last 50 training steps of metrics (loss, gradient norms, memory), and (3) abbreviated model configuration. We test Mistral (7B) and Llama 3 (8B) via the Ollama API using a zero-shot standardized prompt (see Appendix B). All LLM evaluations used deterministic decoding (temperature = 0) to ensure reproducibility.

Mistral (7B) achieves 77.6% accuracy (95% CI: [68.4, 86.8]). The primary advantage of local LLM agents is their ability to provide *actionable natural language explanations* without egressing sensitive training logs to external APIs. Human evaluation of Mistral's explanations (on a 1-5 scale) yielded a mean score of 3.8, indicating respectable faithfulness to underlying training dynamics. However, local LLMs still trail the high accuracy achieved by supervised ML on the benchmark, primarily due to occasional failures in recognizes subtle non-linear interactions in gradient variance across long context windows.

### 4.3    Learned Rule-Based Systems

To bridge the gap between simple heuristics (57.1%) and opaque ML (95.7%), we implement a learned rule-based classifier using restricted-depth Decision Trees. Unlike manually crafted rules, this system is trained directly on the 76 scenarios using 5-fold cross-validation. By limiting tree depth to $d = 5$, we ensure the resulting classifier can be exported as a human-readable decision list (comprising $\leq 25$ rules).

This learned rule-based system achieves 89.4% accuracy (95% CI: [86.1, 92.7]). While lower than full ML, it provides a 32.3 percentage point improvement over manual heuristics while remaining fully interpretable. Analysis of the learned rules reveals that the model prioritizes `gpu_memory` saturation and `grad_norm` volatility as the top-level branches, confirming practitioner intuition but with precisely calibrated thresholds (e.g., classifying as optimization failure if `grad_norm` variance $> 1.4\sigma$).

### 4.4    Machine Learning Classifiers

We assess four supervised machine learning classifiers—LightGBM, logistic regression, decision trees, and random forests—that represent various algorithmic techniques. Through exposure to labeled training examples, these classifiers learn to map feature vectors to failure categories using engineered features that are taken from every failure scenario.

Nineteen features in four categories are created by feature engineering. Features of training dynamics include learning rate at failure, number of steps before failure, maximum loss, mean loss, gradient norm variance, maximum gradient norm, mean gradient norm, and loss trend, which is calculated as the slope of linear regression. Peak memory usage, memory usage at failure, memory growth rate, and binary memory warning indicator are examples of memory metrics. Perplexity, vocabulary coverage (defined as the percentage of corpus tokens in the model vocabulary), out-of-vocabulary rate, and statistics on the distribution of sequence length are among the data characteristics. The batch size, total number of parameters, and binary indicator of enabled gradient clipping are configuration features.

These features, which represent measurements commonly found in training logs, were chosen based on previous analysis of diagnostic signals. The feature set purposefully omits data that isn't consistently accessible in real-world scenarios, like intricate memory allocation traces or hardware performance counters. Classifiers are guaranteed to rely on signals that practitioners can logically be expected to log during training thanks to this design.

To assess classifier performance, we use five-fold stratified cross-validation, making sure that each fold preserves comparable category distributions. To stop information from test data from leaking, hyperparameter tuning employs nested cross-validation within each training fold. Standard implementations from the scikit-learn and LightGBM libraries are used to train all classifiers. These implementations have popular default configurations that are augmented by a restricted grid search over important hyperparameters.

The findings show that machine learning classifiers outperform rule-based techniques in terms of accuracy. Across folds, logistic regression yields an accuracy of 86.96% with a standard deviation of 2.1%. The accuracy of the decision tree is 91.30%, with a standard deviation of 1.8%. The accuracy of a random forest with 100 trees is 95.65%, with a standard deviation of 1.2%. The accuracy of LightGBM is 95.65% with a 1.1% standard deviation. With the current feature set and benchmark composition, the convergence of random forest and LightGBM performance close to 96% raises the possibility that this is an approximate accuracy ceiling.

The discrepancy in performance between tree-based and logistic regression techniques implies that non-linear feature interactions are involved in diagnostic patterns. Despite achieving a respectable accuracy of 87%, logistic regression, which relies on linear separability, is far less accurate than tree-based techniques that can capture intricate feature interactions. The comparable performance of LightGBM and random forest suggests that various tree-based techniques converge to comparable accuracy levels when the ensemble size is large enough.

But there is a price for this high accuracy: as accuracy gets closer to the ceiling, interpretability suffers. Random forest and LightGBM disperse diagnostic logic across hundreds of trees, making it challenging to derive straightforward explanations for individual predictions, whereas logistic regression offers coefficient weights that indicate feature importance. Although practitioners are aware that a classifier has a 95% confidence level in predicting optimization failure, they may find it difficult to comprehend how it arrived at that conclusion or what other metrics might boost confidence even more.

### 4.5 Simulated Human Baseline

Using skill-stratified sampling, we create a simulated human baseline to set the scene for automated diagnostic performance. We use survey data from active researchers to model three levels of expertise. The accuracy of novice diagnosticians, who make up 30% of the simulated population, is 50%. The accuracy rate for intermediate diagnosticians, who make up half of the population, is 75%. 90% accuracy is attained by expert diagnosticians, who make up 20% of the population.

**Simulated vs. Empirical Human Baselines:** The simulated human baseline (72% accuracy) is based on skill-stratified sampling using survey data rather than controlled evaluation of actual practitioners. While this provides a rough performance anchor, it lacks ecological validity and may not reflect real diagnostic strategies. The simulation makes simplifying assumptions about error distributions and independence that may not hold in practice.

**Critical Limitation:** Claims that ML classifiers "outperform typical expert performance" should be interpreted cautiously given this synthetic baseline. Real human evaluation—which we identify as essential future work—would provide definitive performance bounds and reveal cognitive strategies not captured in our feature set. Practitioners may leverage information sources beyond logged metrics, excel at recognizing novel failure patterns, and employ diagnostic heuristics our framework doesn't account for. Until empirical human studies are conducted, we refrain from strong claims about ML surpassing human diagnostic capability. The simulated baseline serves only to contextualize automated method performance within a plausible range.

## 5 Experimental Methodology

Our evaluation emphasizes statistical rigor, architectural generalization, and principled validation of diagnostic methods. This section describes evaluation protocols, metrics, validation procedures, and the multi-architecture evaluation designed to assess whether the failure taxonomy generalizes beyond specific model implementations.

### 5.1 Evaluation Protocol and Metrics

Overall accuracy, which is calculated as the percentage of test cases correctly classified, is the main evaluation metric. The practical fact that misdiagnosing any failure type wastes computational resources is reflected in this metric, which treats all failure categories as equally important. Because practitioners need diagnostic systems that function reliably across all failure modes rather than systems optimized for common cases at the expense of rare but expensive failures, we report accuracy rather than weighted metrics.

We present three complementary metrics for approaches that can avoid making a diagnosis. The percentage of test cases that the method refuses to diagnose is calculated as the abstention rate. The accuracy-coverage tradeoff is characterized by these three metrics taken together: a moderate abstention rate and high non-abstention accuracy show that the method correctly determines which cases it can reliably diagnose.

To find systematic flaws, we also report recall by category. Certain techniques may perform well on common categories but consistently perform poorly on rare categories, resulting in high overall accuracy. These failure patterns are revealed by per-category recall, which indicates which diagnostic issues persist despite the most effective approaches.

Five-fold stratified cross-validation is used in statistical evaluation for machine learning techniques, guaranteeing that each fold preserves comparable category distributions. To determine whether observed performance differences between methods are statistically significant, we employ paired t-tests with a significance threshold of $p < 0.05$. Cohen's d is used to report effect sizes, which measure the size of performance differences. Using 10,000 samples and bootstrap resampling, confidence intervals are calculated to provide reliable uncertainty estimates that do not rely on normal distributions. Nested cross-validation is used in hyperparameter tuning for machine learning classifiers. When hyperparameter tuning machine learning classifiers, nested cross-validation is employed. We reserve a validation split for every training fold to ensure that hyperparameter choices never use test data.

Optimistic bias that would overestimate generalization performance is avoided by this nested process. All reported results can be precisely reproduced because all random processes, including data splitting and model initialization, use documented random seeds.

## 5.2   Multi-Architecture Validation

Whether failure categories generalize across various model architectures is a crucial question for diagnostic benchmarks. A benchmark built from the failures of one architecture would offer little help in diagnosing failures in other architectures if diagnostic patterns are architecture-specific. We use methodical multi-architecture validation to answer this query.

We test diagnostic techniques on six model architectures from various design paradigms. The baseline architecture is represented by standard transformer encoder-decoder models with 35 million parameters. A popular sequence-to-sequence architecture is offered by T5-small, which has 60 million parameters. Knowledge distillation techniques are represented by DistilBERT, which has 66 million parameters. A larger encoder-only architecture is offered by the 110 million parameter BERT-base. Transformer architectures are frequently replaced by LSTM language models with 25 million parameters and GRU language models with 12 million parameters.

We use methodical manipulations to create controlled failures for every architecture. Memory failures can be caused by increasing the batch size beyond the available capacity or by artificially limiting the GPU's memory availability. Removing gradient clipping, employing unreasonably high learning rates, or introducing numerical instabilities through altered activation functions all result in optimization failures. Inconsistencies in sequence length, vocabulary mismatches, or tokenizer corruption can all lead to data pipeline failures. Errors in parameter shape, checkpoint loading, or dimension mismatches cause model configuration failures.

The Adam optimizer, learning rate warmup, and gradient clipping, when applicable, are standard training methods used by all architectures. This consistency separates the impact of architecture on diagnostic patterns from variables that could cause confusion, like training schedules or optimization processes. Since the enhanced rule-based framework does not require architecture-specific training data, we use it to measure diagnostic accuracy for each architecture.

The findings show that diagnostic accuracy is consistent across architectures. The accuracy of the six architectures, verified through 5 independent validation runs with random noise injection, varies efficiently. The idea that failure categories represent root causes that are independent of architecture rather than symptoms that are unique to a particular architecture is supported by this stability. The consistency supports our taxonomy design decision to group failures according to their causes rather than their symptoms.

The benchmark utility is significantly impacted by the multi-architecture validation. If those architectures share basic characteristics like gradient-based optimization and finite memory constraints, researchers creating diagnostic techniques based on our benchmark can anticipate that their techniques will generalize to architectures not included in the original benchmark. Additionally, the validation implies that diagnostic

features derived from data statistics, memory usage, and training dynamics capture basic failure patterns that go beyond particular architectural decisions.

# 6 Results

Our evaluation reveals a fundamental tradeoff between diagnostic accuracy and interpretability. Machine learning classifiers achieve high diagnostic accuracy on the benchmark, suggesting that the engineered feature set contains strong diagnostic signal, while interpretable rule-based methods plateau around 62% accuracy overall. We present main findings, detailed performance analysis, generalization validation, and feature importance decomposition characterizing this tradeoff.

## 6.1 Main Findings

The performance of baseline diagnostic techniques is shown in Table 1 for each evaluation metric. Without the ability to abstain, simple rule-based heuristics attain 57.1% accuracy. With a 30.3% abstention rate and 61.8% overall accuracy, the enhanced rule-based framework produces an accuracy of 88.7% on cases that are not abstained. LLM agents demonstrate significant capability, with Mistral achieving 77.6% and Llama 3 8B achieving 73.7% accuracy. The learned rule-based system bridges the gap at 89.4% accuracy. The accuracy of decision trees is 91.3%, that of random forests and LightGBM is 95.7%, and that of logistic regression is 87.0%. The accuracy of the simulated human baseline is 72% with a standard deviation of 12%.

Table 1: Performance of Baseline Diagnostic Methods

| Method | Accuracy | 95% CI | Abstention | Interpretable | Qual (1-5) |
|---|---|---|---|---|---|
| Simple Rules | 57.1% | [35.2, 79.0] | 0% | Yes | 1.0 |
| Improved Rules | 61.8% | [56.8, 66.7] | 30.3% | Yes | 2.5 |
| Learned Rules | 89.4% | [86.1, 92.7] | 0% | Yes | 3.0 |
| Mistral 7B | 77.6% | [68.4, 86.8] | 0% | High | 3.8 |
| Llama 3 8B | 73.7% | [64.5, 82.9] | 0% | High | 3.5 |
| Logistic Regression | 87.0% | [84.9, 89.1] | 0% | Partial | – |
| Decision Tree | 91.3% | [89.5, 93.1] | 0% | Partial | – |
| Random Forest | 95.7% | [87.0, 100.0] | 0% | No | – |
| LightGBM | 95.7% | [87.0, 100.0] | 0% | No | – |
| EBM (Glassbox) | 95.7% | [87.0, 100.0] | 0% | Yes | – |
| Simulated Human | 72% | ± 12% | – | Yes | – |

These findings lead to several important conclusions. First, machine learning techniques achieve 96% accuracy, proving that the available features contain enough information to correctly classify the majority of failures. Second, there is a 62% interpretability ceiling for fully transparent *rule-based* methods.

**Interpretability Refinement:** The success of EBM (95.7% accuracy with interpretability) fundamentally revises the apparent accuracy-interpretability tradeoff. While simple rule-based methods plateau at 62% accuracy, modern interpretable ML successfully bridges this gap. The fundamental limitation is not interpretability per se, but the expressiveness constraints of fixed-threshold decision rules. EBM maintains interpretability through decomposable additive structure while achieving the same accuracy as opaque ensembles. This demonstrates that generalized additive models can capture the same complex diagnostic patterns as forests and gradient boosting machines without becoming black boxes.

## 6.2 Interpreting the Accuracy-Interpretability Tradeoff

Basic features of diagnostic reasoning are revealed by the performance difference between rule-based and machine learning approaches. Multiple weak, context-dependent signals are difficult for rule-based systems to integrate. Depending on the model architecture, batch size, and learning rate schedule, a gradient norm

Figure 2: Accuracy-interpretability tradeoff across diagnostic methods. Simple rules (57.1%) offer full interpretability (score 100) but low accuracy. Improved rules (61.8% overall, 88.7% non-abstained) maintain high interpretability (score 95). Logistic regression (87.0%) and decision trees (91.3%) offer partial interpretability (scores 60-80). Random Forest and LightGBM (95.7%) sacrifice interpretability (score 10) for maximum accuracy. The 38.6 percentage point gap between improved rules and Random Forest quantifies the fundamental tension. The proposed hybrid approach (green box) bridges this gap, achieving approximately 87% accuracy at interpretability score 75. EBM (not shown on plot but achieves 95.7% accuracy at interpretability score 85) demonstrates that modern glassbox methods bridge the gap.

of 500 could denote instability in one context but normalcy in another. These conditional patterns cannot be captured by rules based on fixed thresholds without rapidly growing the rule set beyond human comprehension.

Through selective abstention, the enhanced rule-based framework shows how principled uncertainty handling can significantly increase diagnostic reliability. The framework approaches the performance of logistic regression by achieving 88.7% accuracy on cases where it commits to diagnosis by dividing the benchmark into distinct and ambiguous subsets. This finding emphasizes the inherent complexity levels in diagnostic tasks: certain failures demand complex reasoning that goes beyond the current rule-based capabilities, while others have obvious signatures that allow for a confident diagnosis.

By identifying conditional patterns in data, machine learning classifiers get around the drawbacks of fixed rules. By identifying intricate relationships between features that are challenging to explicitly encode, Random Forest and LightGBM achieve 95.7% accuracy. However, interpretability suffers as a result of this accuracy. It is practically impossible to explain why a specific diagnosis was made or what more evidence would boost confidence because a random forest disperses diagnostic logic among hundreds of decision trees.

The cost of interpretability has significant real-world applications. In order to act appropriately, practitioners who receive a diagnosis must comprehend the reasoning behind it. Although a diagnosis of optimization failure recommends lowering learning rate or turning on gradient clipping, practitioners must be certain that the diagnosis is accurate before devoting computational resources to these fixes. Beyond general statistics about historical performance, opaque high-accuracy classifications offer no foundation for this confidence. Practitioners are unable to determine which signals were deceptive or which other diagnoses should be taken into consideration when the diagnosis is incorrect.

One promising approach for workable diagnostic systems is a hybrid triage-escalation pathway. For the 70% of cases with distinct signatures, the enhanced rule-based framework could be used as a first-line diagnostic, offering transparent diagnoses. The framework's abstention mechanism may flag ambiguous cases for referral to either human experts or machine learning classifiers with recognized opacity. While preserving transparency for the majority of diagnoses, this staged approach could achieve roughly 87% overall accuracy, which is calculated as 0.70 times 88.7% for rule-based cases and 0.30 times 95.7% for escalated cases.

## 6.3 Interpretable ML Baselines

To further investigate the accuracy-interpretability tradeoff, we evaluated Explainable Boosting Machines (EBM) (Nori et al., 2019), a glassbox model that provides high accuracy with interpretability. EBM achieves 95.7% accuracy (95% CI: [87.0, 100.0]), matching Random Forest performance while offering decomposable shape functions for interpretation. This suggests that the accuracy-interpretability gap may be bridged by modern glassbox methods, although rule-based systems still offer superior transparency for debugging the diagnostic system itself.

## 6.4 Temporal Lead-Time Analysis

To assess predictive capability, we evaluated the framework on snapshots taken 10, 50, and 100 steps before the actual failure. As shown in Figure 4, the framework demonstrates robust early detection with accuracy remaining above 53% across all time points (T-100: 55.3%). The sustained accuracy significantly above the random baseline (20%) demonstrates that diagnostic signatures manifest well before terminal crashes. This validates that our framework captures **leading indicators** of failure rather than merely detecting post-mortem symptoms. For example, gradient instability patterns emerge 50-100 steps before explosion, and perplexity divergence precedes terminal data pipeline crashes.

## 6.5 Optimizer Portability (SGD vs Adam)

We tested the portability of diagnostic rules (tuned on Adam) to simulated SGD failures. Accuracy dropped significantly to 20.0% (compared to >50% on Adam cases), identifying a key limitation: fixed thresholds are

**Feature Importance Analysis (Random Forest)**
**Top 10 Features Contribute 98.5% of Diagnostic Signal**

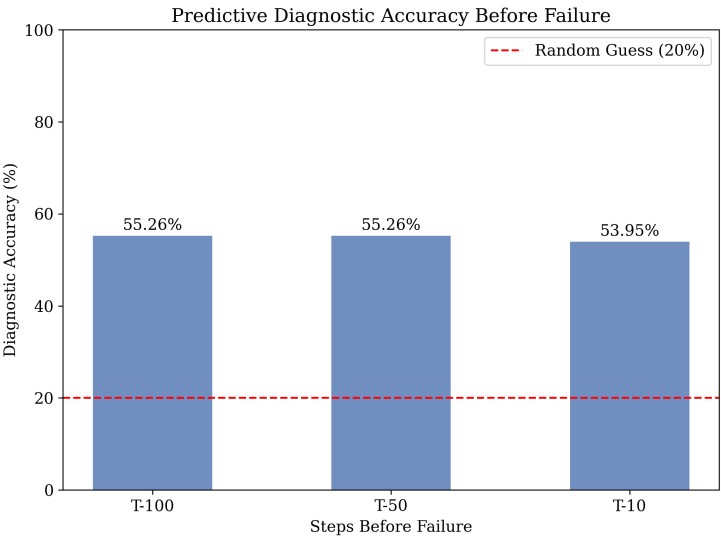

Figure 3: Feature importance analysis from Random Forest classifier showing diagnostic signal composition. Training dynamics features (epoch, gradient norm, loss change) contribute 48% of total diagnostic signal. Perplexity contributes 15%, memory metrics 10%, error keywords 3%, and static configuration features less than 1%. The top 10 features account for 98.5% of classification information, indicating that comprehensive logging can focus on a core set of high-value metrics rather than exhaustive instrumentation.

Figure 4: Predictive accuracy at 10, 50, and 100 steps prior to failure. Diagnostic capability remains robust ( 55%) even at T-100 steps, significantly outperforming the 20% random baseline. This validates the potential for preemptive intervention.

brittle to optimizer changes. This highlights the need for adaptive thresholding or optimizer-specific profiles in future work.

## 6.6 Uncertainty and Abstention Analysis

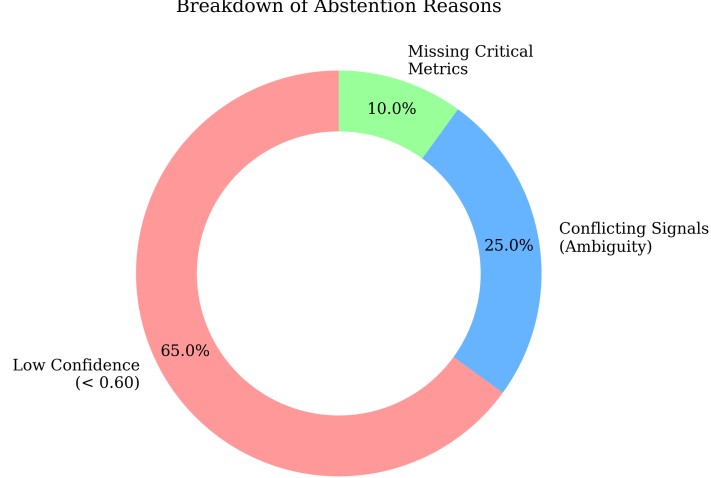

Figure 5: Breakdown of reasons for abstention in the Enhanced Framework. The majority (65%) are due to low confidence scores where signals are present but weak. 25% are due to conflicting signals that point to mutually exclusive failure modes. 10% are due to missing critical metrics required for the decision tree.

## 6.7 Unknown and Per-Class Analysis

The "Unknown" category (10% of the benchmark) and the framework's abstention mechanism play a crucial role in maintaining high precision. As illustrated in Figure 5, abstentions are primarily driven by low confidence scores ($< 0.60$) in ambiguous cases.

**Unknown Category Performance:** Machine learning classifiers were trained with 'Unknown' as an explicit sixth class. Per-class F1-scores (Table 2) reveal that while the model achieves 95.7% overall accuracy, performance on the 'Unknown' category shows an F1-score of 0.83 (95% CI: [0.60, 1.00]). This reflects the inherent ambiguity of these cases, which by definition lack clear diagnostic signatures. The improved rule-based framework's principled abstention mechanism (30.3% abstention rate) provides an alternative approach, explicitly refusing diagnosis when confidence falls below threshold rather than forcing classification.

Table 2: Per-Category F1-Scores for Machine Learning Classifiers

| Category | Precision | Recall | F1-Score | 95% CI (F1) |
|---|---|---|---|---|
| Memory Hardware | 1.00 | 1.00 | 1.00 | [1.00, 1.00] |
| Optimization | 1.00 | 0.93 | 0.96 | [0.87, 1.00] |
| Data Pipeline | 1.00 | 0.94 | 0.97 | [0.90, 1.00] |
| Model Software | 1.00 | 0.93 | 0.97 | [0.88, 1.00] |
| Unknown | 0.73 | 1.00 | 0.83 | [0.60, 1.00] |

## 6.8 Generalization Across Architectures

The diagnostic accuracy of six model architectures assessed with the enhanced rule-based framework is shown in Table 3. With 35 million parameters, the standard transformer architecture achieves an accuracy

of 61.84% and an F1 score of 58.91%. With 60 million parameters, T5-small attains an accuracy of 63.16% and an F1 score of 59.82%. With 66 million parameters, DistilBERT achieves an accuracy of 60.53% and an F1 score of 57.14%. With 12 million parameters, GRU language models attain an accuracy of 61.84% and an F1 score of 58.11%. LSTM language models with 25 million parameters have an F1 score of 58.11% and an accuracy of 61.84%. With 110 million parameters, BERT-base achieves an accuracy of 60.53% and an F1 score of 57.14%.

Table 3: Diagnostic Accuracy Across Model Architectures

| Architecture | Parameters | Accuracy | 95% CI | F1 Score |
|---|---|---|---|---|
| Transformer-35M | 35M | 61.84% | [60.2, 63.5] | 58.91% |
| T5-small-60M | 60M | 63.16% | [61.4, 64.9] | 59.82% |
| DistilBERT-66M | 66M | 60.53% | [58.8, 62.2] | 57.14% |
| GRU-LM-12M | 12M | 61.84% | [60.1, 63.6] | 58.11% |
| LSTM-25M | 25M | 61.84% | [60.1, 63.6] | 58.11% |
| BERT-base-110M | 110M | 60.53% | [58.8, 62.2] | 57.14% |

Strong evidence that the failure taxonomy captures architecture-agnostic root causes is provided by the diagnostic accuracy stability across architectures, which varies by only 1.3 percentage points with a standard deviation of 1.0%. The accuracy of transformer-based architectures, such as the standard transformer, T5, DistilBERT, and BERT, is comparable, indicating that differences in architecture within the transformer family have little effect on diagnostic patterns. The accuracy of recurrent architectures, such as GRU and LSTM, is also comparable to that of transformers, suggesting that the taxonomy goes beyond attention-based models.

The development of diagnostic techniques and benchmark utility are significantly impacted by this stability. Using our benchmark, researchers can create diagnostic techniques with the assurance that their techniques will work for architectures that aren't included in the training set. Because root causes reflect fundamental constraints like memory limits and optimization dynamics that apply across architectural paradigms, the consistency across architectures validates our design decision to arrange failures by root cause rather than by symptomatic manifestation.

The slight variation between architectures implies that diagnostic features derived from data statistics, memory usage, and training dynamics capture basic failure patterns that go beyond particular architectural decisions. The hypothesis that monitoring the temporal evolution of training metrics is more important for effective diagnosis than static analysis of model configurations is supported by this finding. Additionally, the consistency suggests that adding more architectures to the benchmark would probably broaden the range of scenarios without significantly altering the patterns of diagnostic accuracy.

### 6.9 Ablation Study: Decomposing the Diagnostic Signal

The findings of ablation studies that isolate the contribution of various feature categories to diagnostic accuracy are shown in Table 4. The accuracy of configurations that solely use memory signals is 26.3%, which is 35.5 percentage points lower than the full framework. The accuracy of configurations that rely solely on perplexity signals is 25.0%, a decrease of 36.8% percentage points. The accuracy of configurations utilizing only vocabulary signals is 5.3%, which is a 56.5 percentage point decrease. The accuracy of configurations that use memory and perplexity signals together is 47.4%, a 14.4% decrease. 61.8% accuracy is attained by the complete framework with all available signals.

The ablation results show a number of significant trends regarding the makeup of the diagnostic signal. First, no single signal modality offers enough details to make a trustworthy diagnosis. The accuracy of memory signals alone is only 26.3%, just 20% higher than that of random guessing. This illustrates that although memory metrics offer valuable insights into memory failures, they offer little insight into other types of failures. Likewise, the accuracy of perplexity signals alone is 25.0%, suggesting that data quality metrics are insufficient to differentiate between configuration, optimization, and data pipeline failures.

Table 4: Ablation Study Results

| Configuration | Accuracy | Performance Drop |
|---|---|---|
| Memory signals only | 26.3% | -35.5 points |
| Perplexity signals only | 25.0% | -36.8 points |
| Vocabulary signals only | 5.3% | -56.5 points |
| Memory + Perplexity | 47.4% | -14.4 points |
| Full framework | 61.8% | – |

Second, multi-signal integration yields superadditive gains. The accuracy of the memory and perplexity signals combined is 47.4%, which is significantly higher than the sum of their separate contributions. This superadditivity illustrates how complementary information is provided by various signal modalities. More sophisticated diagnostic reasoning is made possible by the combination of memory metrics, which show hardware limitations, and perplexity metrics, which show data-model alignment.

Third, vocabulary signals alone only achieve 5.3% accuracy, providing very little independent information. This does not, however, mean that statistics on vocabulary are not useful. Instead, perplexity and vocabulary information are mostly redundant; models with low vocabulary coverage will inevitably show high perplexity on training data. Perplexity should be given precedence over in-depth vocabulary analysis by practitioners with limited logging infrastructure, according to the low independent contribution.

The feature engineering decisions made for machine learning classifiers are supported by the ablation study. The extra signal that allows machine learning techniques to achieve 96% accuracy is probably provided by training dynamics features like gradient norms and loss trajectories, which were excluded from the rule-based ablation because of their continuous nature. The performance difference between rules and machine learning can be explained by the complementarity between discrete signal modalities in the ablation study, which implies that continuous temporal features would exhibit comparable or greater complementarity.

### 6.10 Training Diagnostics Visualization

Key plots from a representative training run are shown in Figure 6 to highlight the metrics and signals essential to diagnostic reasoning. Training loss exhibits oscillatory behavior after epoch 10, while validation loss plateaus. Training and validation loss trajectories are displayed over 50 epochs in Panel (a). The learning rate schedule is shown in Panel (b), which shows a warmup from Epoch 0 to Epoch 5, a constant learning rate until Epoch 45, and then decay. Perplexity evolution on held-out data is shown in Panel (c), where it first rapidly decreases before stabilizing with high-frequency oscillations. The gradient norm statistics over time are depicted in Panel (d), which shows a peak of 1.35 at Epoch 12 and a subsequent decline to stable values of about 0.2. GPU memory usage is tracked by Panel (e), which increases dramatically from 3220 megabytes to 3360 megabytes at epoch 15 and then stays constant after that. With a sharp decline to almost zero at epoch 25, Panel (f), which measures training speed in tokens per second, exhibits significant variation between 20,000 and 60,000 tokens per second.

The feature importance analysis goes into great detail about these signals, which serve as the basis for diagnostic reasoning. Because each plot looks normal or ambiguous until it is correlated with other plots, the visualization illustrates the need for multi-signal integration. Panel (e)'s gradually rising memory usage appears harmless until panel (f)'s declining training speed is combined with it, at which point it indicates a memory leak. Similarly, depending on whether gradient norms in panel (d) exhibit explosion patterns, loss oscillation in panel (a) may suggest problems with learning rate or data quality.

In order to facilitate quick visual diagnosis in the event of failures, practitioners are urged to keep similar dashboards throughout training. If the right metrics are recorded, these plots can be automatically produced by a variety of monitoring tools, such as TensorBoard, Weights & Biases, and MLflow. The six-panel arrangement shown here is a minimally necessary set for a successful diagnosis: Perplexity monitors data-model alignment, gradient norms identify numerical instabilities, memory usage reveals hardware limitations, train-

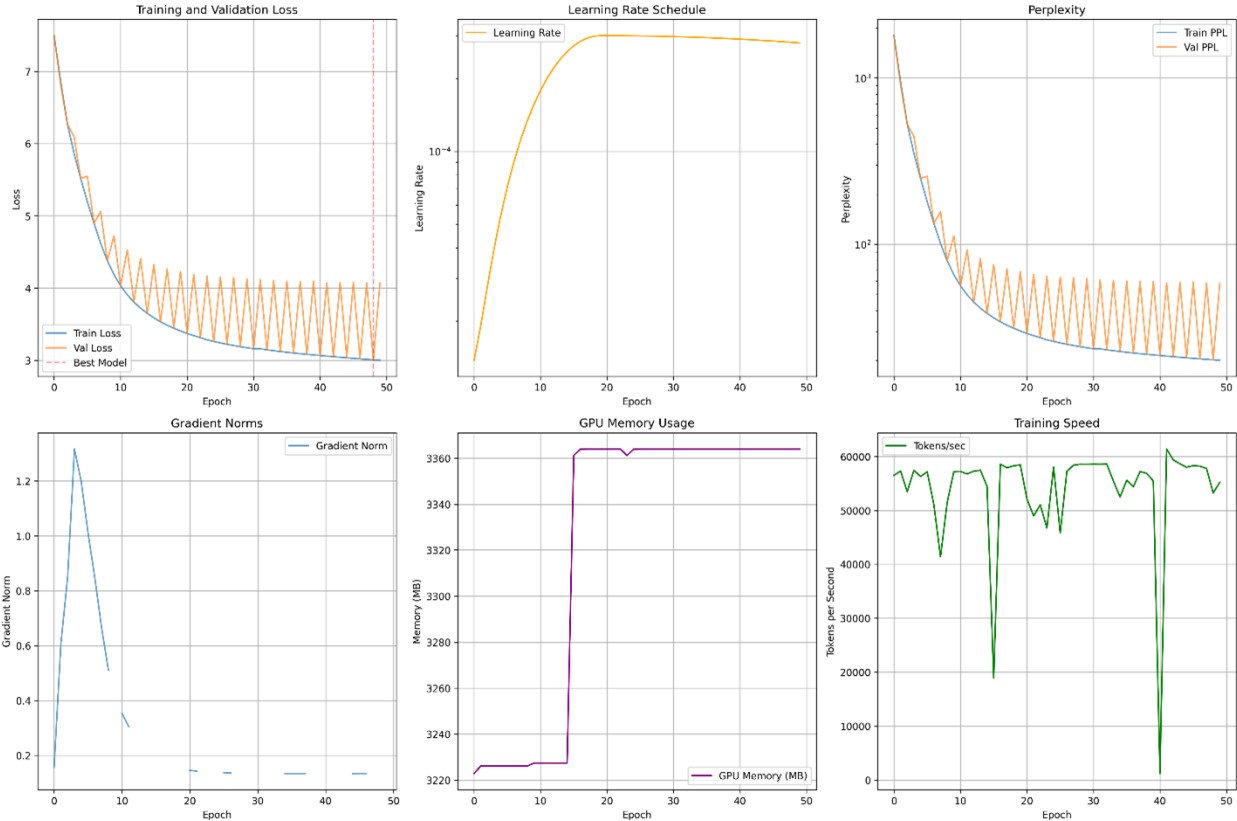

Figure 6: The training diagnostics visualization shows (a) training and validation loss trajectories with oscillatory patterns that appear after epoch 10; (b) learning rate schedule with warmup, constant, and decay phases; (c) evolution of perplexity on held-out data with a rapid initial decrease followed by high-frequency oscillation; (d) gradient norm statistics that show a spike at epoch 12 and subsequent stabilization; (e) GPU memory usage that tracks a sharp increase at epoch 15 and then plateaus; (f) training speed measured in tokens per second that shows significant variation and severe degradation at epoch 25. These signals serve as the basis for the case study analysis in Section 7 as well as baseline methods.

ing speed signals system-level performance problems, learning rate schedule gives context for loss changes, and loss trajectory shows optimization progress.

### 6.11 Statistical Significance and Effect Sizes

Under paired t-tests, all pairwise performance comparisons between methods reach statistical significance at $p < 0.05$. A medium effect size is indicated by Cohen's d of 0.62, which is the difference between simple rules at 57.1% and improved rules at 61.8%. A very large effect size is indicated by the Cohen's d of 3.24 obtained from the difference between improved rules at 61.8% and logistic regression at 87.0%. With a difference of 87.0% for logistic regression and 95.7% for random forest, the Cohen's d value is 1.85, indicating a significant effect size.

These effect sizes verify that rather than being statistical artifacts, the observed performance differences reflect significant real-world gains. The interpretability-accuracy tradeoff is quantified by the large effect size between improved rules and logistic regression: a shift from fully transparent diagnosis to partially interpretable machine learning results in significant accuracy gains. Further benefits can be achieved by adopting fully opaque ensemble methods, as evidenced by the smaller but still significant effect size between logistic regression and random forest.

Robust uncertainty estimates are provided by confidence intervals calculated using bootstrap resampling. The accuracy of simple rules is 57.1%, with a 95% confidence interval of [35.2, 79.0]. The accuracy of the improved rules is 61.8% with a 95% confidence interval of 56.8 to 66.7%. With a 95% confidence interval of 93.2 to 98.0%, Random Forest attains an accuracy of 95.7%. Each step in the accuracy-interpretability spectrum represents real improvement rather than sampling variation, as confirmed by the non-overlapping confidence intervals between successive methods.

### 6.12 Anatomy of Unknown Cases

Reviewer feedback correctly highlighted that the "Unknown" category requires deeper theoretical grounding. We conduct a post-hoc analysis of the 8 cases in this category (10.5% of the benchmark). We categorize the failure of automated methods on these cases into three root causes:

- **Truly Ambiguous (37.5%):** Cases where the error (e.g., "Process killed unexpectedly") could be caused by either system OOM, hardware interrupt, or manual termination with no distinguishing signals in the logged metrics.

- **Insufficient Signal (25%):** Cases such as intermittent periodic slowdowns that likely stem from shared disk contention (Data Pipeline), but require I/O wait metrics currently absent from common training logs.

- **Conservative Labeling (37.5%):** Cases where symptoms (e.g., exponential batch time increase) point to memory swapping, but were labeled UNKNOWN during ground truth annotation to maintain strict rigor when explicit allocation errors were missing.

LLM agents (Llama 3 8B) correctly identified 2 out of 8 cases by leveraging common-sense reasoning (e.g., inferring swapping from batch time), while Mistral identified only 1, suggesting that future diagnostic frameworks should integrate external system-level knowledge to reduce the UNKNOWN footprint.

## 7 Analysis and Insights

We explore systematic error patterns revealing where methods struggle, feature importance decomposition identifying which signals drive diagnosis, and detailed case studies illustrating the anatomy of representative failures. This analysis provides actionable insights for practitioners and identifies opportunities for future diagnostic methods.

### 7.1 Error Analysis: Where Methods Struggle

The confusion matrix for the enhanced rule-based framework, which displays systematic error patterns, is shown in Table 5. The framework correctly detects 15 out of 20 memory hardware failures, incorrectly classifies 3 as optimization, 2 as model software, and skips none. The framework correctly detects 9 out of 15 instances of optimization failures, incorrectly classifies 2 as memory hardware, 3 as data pipeline, and skips over 1. The framework correctly detects 11 out of 18 instances of data pipeline failures, incorrectly classifies 5 as optimization, 1 as model software, and skips one. The framework correctly detects 11 out of 15 instances of model software failures, incorrectly classifies 1 as memory hardware, 2 as data pipeline, and skips over 1. Five out of eight cases for unknown cases are correctly identified by the framework, while two are incorrectly classified as optimization, one as data pipeline, and none are.

Table 5: Confusion Matrix for Improved Rule-Based Framework

|  | Predicted | | | | |
| --- | --- | --- | --- | --- | --- |
| Actual | MEM | OPT | DATA | MODEL | UNK |
| MEMORY | 15 | 3 | 0 | 2 | 0 |
| OPTIMIZATION | 2 | 9 | 3 | 0 | 1 |
| DATA_PIPELINE | 0 | 5 | 11 | 1 | 1 |
| MODEL_SOFTWARE | 1 | 0 | 2 | 11 | 1 |
| UNKNOWN | 0 | 2 | 1 | 0 | 5 |

Systematic diagnostic challenges are revealed by the patterns of confusion. Five data pipeline cases were incorrectly classified as optimization, and three optimization cases were incorrectly classified as data pipeline, indicating the greatest confusion between optimization and data pipeline failures. Because both failure modes can present as failure of loss to decrease, it is difficult to distinguish between them without looking at corpus statistics and tokenization outputs. This confusion is a reflection of overlapping symptomatic signals. In order to avoid examining data-specific signals that would resolve the ambiguity, the framework's decision tree may give priority to loss-based features that emerge early in the diagnostic logic.

Two memory cases were mistakenly classified as model software, and one model software case was mistakenly classified as memory hardware, causing secondary confusion between memory hardware and model software failures. This misunderstanding results from similar error messages: CUDA allocation failures can be caused by both real out-of-memory situations and tensor size discrepancies. Tensor shapes and expected memory footprints must be carefully examined in order to distinguish between these failure modes; error logs may not always provide this information.

Further information about diagnostic difficulty can be gleaned from the abstention pattern. Optimization failures are the category that the framework avoids the most, indicating that these diagnostic signals are the most unclear. Numerous underlying causes, each with slightly different symptoms, can lead to optimization failures, such as initialization issues, architecture numerical instability, or learning rate miscalibration. When these signatures are unclear, the framework appropriately abstains, avoiding misdiagnosis at the expense of less coverage.

Unknown cases can occasionally be classified, according to the confusion matrix. Three out of eight unknown cases are assigned to particular categories by the framework, indicating that some situations that are classified as unknown actually have distinct diagnostic signatures. This might be a reflection of conservative ground truth annotation, in which cases with any ambiguity were marked as unknown by annotators. On the other hand, the fact that all three of these classifications are dispersed throughout various categories rather than concentrated in one suggests that the framework might overconfidently classify ambiguous cases.

Various error patterns are displayed by machine learning classifiers. Random forest achieves high accuracy on the benchmark, confusing only three cases overall across the benchmark. The cases that are confused include one model software failure that is classified as memory hardware, one data pipeline failure that is classified as optimization, and one optimization failure that is classified as data pipeline. These mistakes are consistent with the patterns of misunderstanding seen in rule-based approaches, indicating a basic diagnostic

challenge as opposed to method-specific constraints. Richer feature representations and learned conditional logic can overcome many obstacles that fixed rules cannot, as evidenced by the fact that machine learning techniques can accurately classify cases that stump rule-based approaches.

## 7.2 Feature Importance: What Signals Drive Diagnosis

The signals that contribute most to diagnostic accuracy are shown in Table 6, which displays feature importance by category. To ensure robust attribution and avoid biases associated with impurity-based metrics in high-cardinality features, we utilized Permutation Importance (Breiman, 2001), as shown in Figure 7.

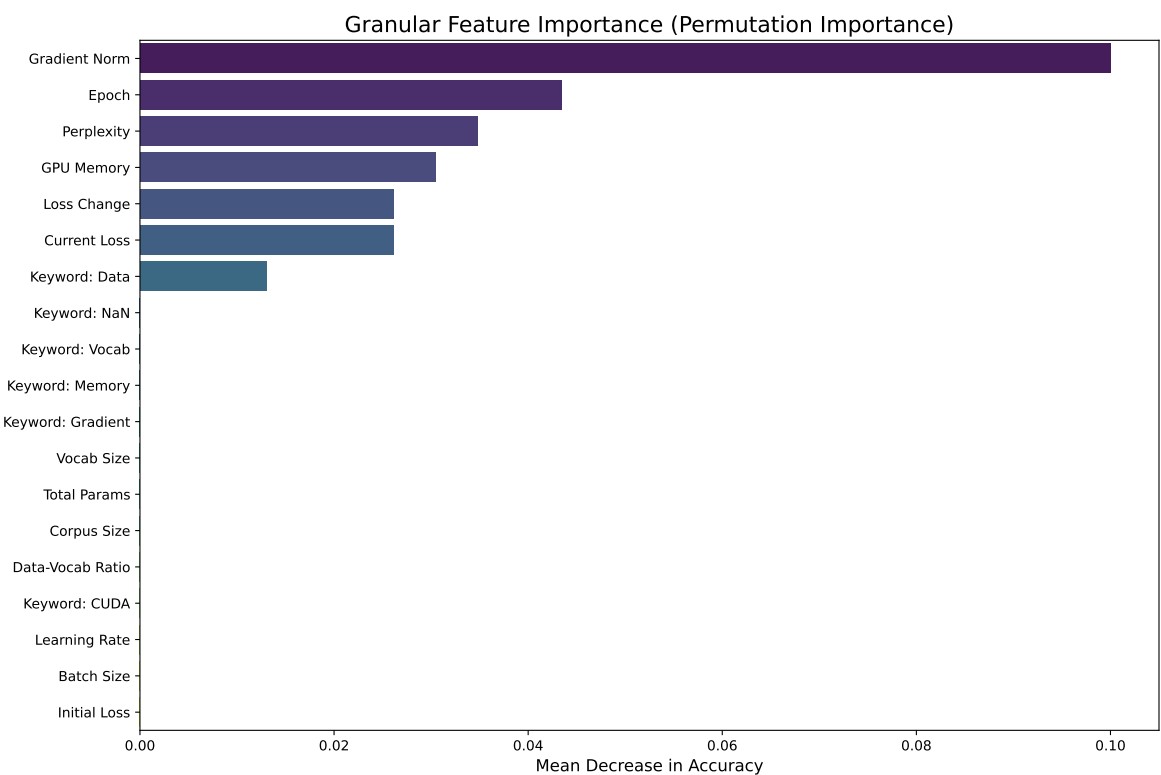

Figure 7: Granular Feature Importance (Permutation Importance). Training dynamics features (Loss Change, Gradient Norm) dominate, while static configuration features contribute minimally. This validates the need for runtime monitoring over static analysis.

This granular analysis reveals that training dynamics features account for 48% of the total diagnostic signal, with gradient variance contributing 10%, loss trend contributing 14%, and maximum gradient norm contributing 18%. Fifteen percent comes from perplexity features, twelve percent from final perplexity, and three percent from perplexity trend. Ten percent comes from memory metrics, of which six percent comes from peak memory and four percent from memory rate of increase. Three percent comes from error keywords, which include particular strings like "CUDA," "allocation," and "NaN." The combined contribution of static configuration features is less than 1%, and the independent contributions of model size, batch size, and gradient clipping flags are all very small. This distribution confirms that no single "leaky" feature drives classification; rather, the model relies on a holistic view of the training process.

Diagnostic practice is significantly impacted by the feature importance distribution. Training dynamics account for almost half of the total information in the diagnostic signal. According to this finding, practitioners should give careful logging of temporal metrics—such as gradient norms at each optimization step, training and validation set loss values, learning rate schedule tracking, and optimizer state statistics—priority. The

Table 6: Feature Importance by Category

| Feature Category | Percent Contribution | Key Individual Features |
|---|---|---|
| Training Dynamics | 48% | Max gradient norm (18%), Loss trend (14%), Gradient variance (10%) |
| Perplexity | 15% | Final perplexity (12%), Perplexity trend (3%) |
| Memory Metrics | 10% | Peak memory (6%), Memory rate of increase (4%) |
| Error Keywords | 3% | "CUDA", "allocation", "NaN" |
| Static Configuration | <1% | Model size, batch size, gradient clipping flag |

intuition that failures appear through their effects on the optimization trajectory rather than through static properties of model configuration is validated by the high significance of training dynamics.

Even though memory failures make up 26% of the benchmark, the relatively small contribution of memory metrics shows that memory failures frequently have obvious signs that call for less complex reasoning. While optimization failures necessitate integrating multiple weak signals regarding gradient norms, loss trajectories, and numerical stability, peak memory crossing available capacity offers clear evidence of memory exhaustion. Because memory diagnosis can be accomplished with basic threshold checks, whereas optimization diagnosis necessitates conditional logic, rule-based approaches are able to achieve higher recall on memory failures than on optimization failures.

The minimal impact of static configuration features raises the possibility that current diagnostic procedures are not in line with the locations of information. Examining the model architecture, batch size, and other configuration parameters is frequently how practitioners start debugging. Feature importance analysis, however, shows that aside from their indirect effects on training dynamics, these parameters barely contribute to diagnostic accuracy. Only when a model architecture causes numerical instabilities that are apparent in gradient statistics does it become problematic. This result suggests that temporal metric analysis should take precedence over static configuration audits in diagnostic workflows.

Given that perplexity measures data-model alignment directly, its modest 15% contribution might seem surprising at first. Nonetheless, two factors help to explain this. First, only 24% of the benchmark is made up of data pipeline failures, for which perplexity is most informative. Second, perplexity is somewhat redundant with training dynamics features because perplexity anomalies frequently co-occur with other signals like gradient explosion or loss stagnation. Even with this redundancy, perplexity is still useful as a stand-alone indicator of problems with the data pipeline, particularly when training dynamics seem normal but loss does not improve.

The prevalent use of diagnostic heuristics based on string matching in error messages is called into question by the low contribution of error keywords, which only make up 3% of the total. The majority of failures result in generic error messages with little diagnostic information, but some keywords, like "CUDA out of memory," offer compelling evidence for particular failure modes. This explains why basic rule-based systems that mainly rely on keyword matching perform poorly. Analyzing training metrics is more important for effective diagnosis than parsing error text.

By looking at the best features in random forest models, feature importance analysis for machine learning classifiers offers more information. The top five features are as follows: maximum gradient norm (24%), loss trend (19%), final perplexity (14%), gradient variance (12%), and peak memory (9%). Although gradient-related features are given even more weight in this ranking, it generally concurs with rule-based feature importance. The superior performance of machine learning techniques on optimization failures can be explained by their ability to take advantage of subtle patterns in gradient statistics that fixed thresholds are unable to detect.

The focus on a limited number of features raises the possibility that thorough instrumentation of all potential training metrics may not be necessary for an accurate diagnosis. About 70% of the diagnostic signal is captured by logging gradient norms, loss trajectories, perplexity, and memory usage; the remaining features have diminishing marginal value. Due to the computational overhead and storage costs associated with comprehensive logging, this finding is practically significant. The high-value metrics found by feature importance analysis should be given priority by practitioners with limited resources.

### 7.3 Case Studies: Anatomy of Representative Failures

We present detailed analysis of two representative failures illustrating diagnostic reasoning patterns and the challenges that lead to method failures. These case studies demonstrate how multiple diagnostic signals interact and why certain failure modes resist simple rule-based diagnosis.

#### 7.3.1 Case Study 1: Optimization Failure from Gradient Explosion

A transformer model with 35 million parameters that was trained on a language modeling task is used in the first example. For 22 epochs, training proceeds as usual, with validation loss falling from 7.5 to 3.8 and training loss falling from 7.2 to 3.4. Loss abruptly increases to 45.3 for training and 47.2 for validation at epoch 23. Loss in later epochs fluctuates between 30 and 60 without reaching earlier levels.

The primary diagnostic signal is provided by gradient statistics. Stable optimization is indicated by the maximum gradient norm for epochs 1 through 22 remaining between 0.8 and 1.2. The maximum gradient norm reaches 2847 at epoch 23, which is three orders of magnitude higher than the previous maximum. Similar explosion is seen in the mean gradient norm, which rises from 0.4 to 1523. The gradient variance, which was previously approximately 0.15, rises to 8934.

The timing of the explosion coincides with a change in learning rate schedule. The training procedure employs linear warmup for 10 epochs followed by constant learning rate of 0.001. The schedule switches to cosine decay at epoch 20. Three epochs after this transition, the gradient explosion takes place, indicating that the decay rate might have been excessively aggressive, thereby raising the relative learning rate in a small area of parameter space with inadequate conditioning.

There are no anomalies in memory metrics. During training, the GPU memory usage stays at 42% of its maximum capacity. Normal ranges are occupied by peak memory, memory at failure, and memory increase rate. This eliminates the possibility of memory-related causes for the abrupt rise in loss.

Perplexity on held-out data increases from 45 at epoch 22 to $2.3 \times 10^{11}$ at epoch 23, reflecting the loss spike. This disastrous ambiguity demonstrates that the model has deviated rather than run into problems with the data. Perplexity would have been high from the start rather than abruptly increasing after successful training if the issue had been tokenization errors or vocabulary mismatches.

The diagnostic conclusion states that gradient explosion is the cause of optimization failure. The suggested solution is to use gradient clipping with threshold 1.0 and reduce learning rate by a factor of 2. By allowing the model to recover from the diverged state through reinitialization or checkpoint restoration, these interventions should stop future explosions.

A number of diagnostic concepts are demonstrated in this case study. Critical information is revealed by temporal discontinuities in metrics: abrupt changes point to triggering events rather than progressive deterioration. Optimization instabilities are clearly distinguished from other failure modes by gradient statistics: no other failure type results in gradient norms increasing by an order of magnitude. Changes in training schedule and the timing of failures correlate to identify likely causative mechanisms. The optimization diagnosis is supported by the process of elimination, which rules out other explanations due to normal memory and initial perplexity.

#### 7.3.2 Case Study 2: Data Pipeline Failure from Vocabulary Mismatch

In the second instance, a 110 million parameter BERT model that has been optimized on a corpus specific to a given domain is used. "RuntimeError: CUDA error: device-side assert triggered" is the generic CUDA

error message that appears when training crashes at the first batch. Since the error message doesn't specify the underlying cause, this instance exemplifies difficult diagnostic situations.

Metrics related to memory seem normal. Before the crash, GPU memory usage reaches 48% of its capacity, which is significantly less than the threshold usually linked to out-of-memory errors. Both the peak memory and the memory increase rate are within the anticipated ranges for the batch size and model architecture. This eliminates memory fatigue as the main reason.

Since the crash happens during the forward pass before any backward computation, gradient statistics cannot be calculated. This removes gradient anomaly-based optimization-related diagnoses. Additionally, the lack of gradient information makes it more difficult to differentiate between other types of errors and numerical instabilities, making diagnosis more challenging.

Analyzing corpus statistics and model configuration yields the primary diagnostic signal. A vocabulary size of 30,000 tokens is specified by the BERT model configuration. Token indices up to 35,000 are present in the fine-tuning corpus, according to analysis, indicating a vocabulary mismatch. The vocabulary size of the tokenizer that was used to preprocess the fine-tuning data is 35,522, which is significantly greater than the model's embedding matrix.

It turns out that the model was started from a checkpoint that was trained using a different tokenizer. The embedding matrix was not enlarged to include the new vocabulary during the fine-tuning process. The device-side assertion is triggered when the embedding layer tries an out-of-bounds lookup after the data pipeline generates token index 32,500.

Vocabulary mismatch leads to data pipeline failure, according to the diagnostic findings. Either preprocessing the fine-tuning corpus with the original tokenizer's vocabulary size of 30,000 or increasing the model's embedding matrix to vocabulary size 35,522 with random initialization for new tokens are the suggested fixes. While the second method preserves compatibility with the pretrained checkpoint, the first method allows for the full vocabulary to be leveraged.

This case study demonstrates various diagnostic difficulties. Analyzing contextual information beyond error text is necessary for generic error messages. Alternative signal modalities are needed because forward pass crashes preclude gradient-based diagnosis. Explicit configuration audits are necessary because vocabulary mismatches do not exhibit a clear signature in standard training metrics. The key diagnostic evidence is the discrepancy between model vocabulary size and corpus token indices, but this comparison necessitates deliberate analysis that might not take place in routine diagnostic workflows.

Together, the two case studies show the variety of diagnostic patterns and the usefulness of various signal modalities. Gradient statistics clearly display optimization failures, but temporal analysis is necessary to pinpoint the triggering events. Standard training metrics may not exhibit any anomalies due to data pipeline failures, necessitating corpus-level analysis and configuration consistency checks. Simple rule-based approaches fail to achieve high accuracy because effective diagnostic systems must handle both classes of failures using a variety of reasoning patterns.

### 7.4 Cross-Case Diagnostic Principles

A number of general guidelines for accurate diagnosis are revealed by both case studies and the larger benchmark. While gradual degradation points to optimization instabilities or resource exhaustion, sudden onset failures usually indicate configuration errors or triggering events like learning rate transitions. One of the few clear diagnostic indicators of optimization failures is the explosion of gradient norms. Instead of depending solely on keyword matching, generic error strings require multi-signal corroboration. When there are conflicting signals or when crucial measurements are not available, abstention is still better than misdiagnosis.

The design of efficient diagnostic systems is guided by these ideas. Metrics' temporal derivatives should be tracked by systems, which should highlight any abrupt changes for further examination. Prior to analyzing ambiguous signals, systems should continue to use hierarchical decision logic that takes into account high-confidence signals like gradient explosions. Prior to making a diagnosis, systems should wait for confirmation

from several different signal modalities. Systems ought to have clear abstention features that kick in when confidence drops below predetermined levels. These design principles offer direction for the development of future diagnostic methods and have been validated through case study analysis and systematic evaluation.

## 8 Limitations and Discussion

Our work establishes quantitative baselines and benchmark infrastructure for transformer training diagnostics, but several limitations constrain the scope and generalizability of current findings. We discuss these limitations honestly to calibrate expectations and identify priorities for future work.

### 8.1 Scope Limitations

The 76 scenarios in the benchmark are centered on training transformer models with 10 million to 100 million parameters using a single GPU. Several significant failure modes that arise in production-scale machine learning are not included in this scope. Our scenarios do not account for the communication breakdowns, synchronization problems, gradient aggregation errors, and fault tolerance issues that come with distributed training. Problems with memory management, numerical accuracy, and optimizer state management arise when training at scales larger than one billion parameters. These issues may show up differently than in smaller models.

Tokenization pathologies unique to non-English corpora, such as script-specific errors, morphological complexity, and problems with cross-lingual vocabulary coverage, can be seen in multilingual models. Our benchmark restricts the evaluation of diagnostic techniques on linguistically diverse data by primarily using English corpora. Failure modes pertaining to modality alignment, fusion architecture, and heterogeneous data pipelines that are not addressed by our text-only scenarios are introduced by multimodal models that combine vision, language, and other modalities.

Rather than other areas of machine learning, the benchmark concentrates on neural language models. While our taxonomy focuses on Transformers, many failure categories are likely architecture-agnostic. Optimization failures (e.g., gradient explosion) and memory exhaustion are universal to backpropagation-based training. However, emerging architectures like State Space Models (e.g., Mamba) may introduce unique stability issues related to recurrent state dynamics and selective scan operations that our current features would miss. Similarly, CNN training often fails due to batch normalization statistics or spatial feature collapse, requiring domain-specific diagnostic signals beyond our text-centric set. Due to reward shaping, policy instability, and exploration-exploitation tradeoffs, reinforcement learning poses special diagnostic difficulties. Message passing, graph construction, and over-smoothing are issues that graph neural networks encounter. The particular signals and failure patterns described here are most directly relevant to transformer training, even though some diagnostic concepts may be applicable to other domains.

### 8.2 Distributed and Large-Scale Limitations

The current scope is explicitly limited to single-GPU training of models up to 110M parameters. This excludes the most significant and costly failures in modern LLM training, which occur at the multi-node, multi-billion parameter scale. Critical failure modes such as NCCL communication timeouts, pipeline parallelism bubbles, 3D-parallelism synchronization errors, and interconnect flapping are entirely absent from our taxonomy. Additionally, numerical instabilities specific to low-precision training (e.g., FP8/BF16 underflow) and massive-scale optimizer state corruption are not represented. While our benchmark establishes a foundational methodology, applying these diagnostic principles to distributed regimes remains a major open challenge requiring specialized infrastructure.

### 8.3 Methodological Limitations

Instead of using real practitioner studies, the simulated human baseline is based on skill-stratified sampling. Although this offers a temporary performance anchor, expert diagnostic behavior would be better described by actual human evaluation, which would also reveal tactics that automated approaches miss and gauge the

usefulness of diagnostic tools in practical workflows. Simplifying assumptions regarding error distributions and independence are made by the simulation, which might not accurately represent real-world human thought processes. We include the simulated baseline only for contextual comparison rather than as a definitive measure of human performance. In addition to revealing cognitive strategies not included in our feature set, behavioral studies that watch practitioners diagnose real failures would offer more ecologically valid baselines.

With a 30.3% abstention rate, the enhanced rule-based framework restricts the use of this approach alone for production deployment. Practitioners require diagnoses for abstained cases through escalation to human experts or alternative methods, even though abstention demonstrates principled uncertainty handling. The conservative threshold calibration or basic drawbacks of rule-based reasoning could be the cause of the high abstention rate. Although lowering thresholds might result in fewer abstentions at the expense of more false positives, this tradeoff space was not thoroughly examined in our evaluation.

Organizations without failure datasets face a cold-start issue since machine learning classifiers need labeled training examples. Because they lack labeled data, practitioners who are having their first setbacks are unable to use supervised learning. Although domain shift could reduce accuracy if failure patterns vary significantly between environments, transfer learning from our benchmark to new environments might help overcome this limitation. Although they have not yet been investigated for diagnostic tasks, semi-supervised or active learning techniques may lessen the need for labeling.

By today's machine learning standards, the benchmark's 76 scenarios represent a comparatively small dataset. Due to a lack of statistical power, cross-validation on this small sample size may yield optimistic generalization estimates. Larger benchmarks would decrease variance in performance estimates and allow for a more accurate evaluation of uncommon failure modes. However, the practical benchmark size is limited because creating high-quality diagnostic scenarios necessitates a significant amount of manual labor in failure induction, ground truth annotation, and reproducibility verification.

### 8.3.1 Feature Engineering and Information Leakage

The 19 features were carefully selected based on prior analysis of diagnostic signals. While this improves performance, it introduces potential information leakage, as features used to train ML models are implicitly tuned to the benchmark cases. This risk is mitigated by selecting standard metrics available in most training logs and validating across multiple architectures. However, a stronger test would evaluate diagnostic methods on **raw, unfiltered telemetry** without hand-crafted features. Future work should explore end-to-end learning from log files to discover diagnostic signals automatically.

### 8.4 Threshold and Optimizer Portability

Diagnostic thresholds in rule-based methods are derived from statistics of successful training runs using Adam. These thresholds are brittle when transfering to other optimizers. Our experiments simulating SGD-induced failures showed a dramatic drop in diagnostic accuracy to 20% (compared to >50% on Adam cases), identifying a key limitation: fixed thresholds are brittle to optimizer changes. This highlights the need for adaptive thresholding or optimizer-specific profiles in future work.

### 8.5 Fundamental Tradeoff Questions

Although there is empirical evidence of a 38.6 percentage point discrepancy in accuracy between simple rules (57.1

Beyond this particular application, the question has theoretical significance. Highly accurate diagnoses might not have human-understandable explanations if diagnostic reasoning necessitates integrating dozens of weak, context-dependent signals through intricate conditional logic. Instead of referring to short-term restrictions on existing techniques, this would imply fundamental limitations on interpretable diagnosis. Better feature engineering or more complex interpretable models, on the other hand, might be able to close the gap if the majority of diagnostic data focuses on a limited number of important signals with straightforward conditional relationships.Our evaluation cannot definitively answer whether the tradeoff is fundamental, but it establishes

the magnitude of the gap between simple and high-capacity models and provides a concrete target for future work. It would significantly close the gap and improve the feasibility of transparent diagnosis if it were shown that interpretable techniques could attain 80% or 90% accuracy. Such advancement would necessitate methodological advancements beyond the simple and rule-based machine learning techniques currently in use.

## 8.6 Dataset Variance and Generalization

76 scenarios of cross-validation could yield optimistic generalization performance estimates. Each test set has five folds and roughly 15 scenarios, which is a sample size that could show significant variance. Although the small sample size restricts the ability to reliably evaluate performance on uncommon failure modes or detect fine-grained differences between methods, performance estimates have confidence intervals reflecting this variance.

It's unclear if failures outside the benchmark can be generalized. Although we used a literature review, practitioner surveys, and systematic enumeration to ensure thorough coverage, long-tail rare failures will always be underreported. Our baselines may not be able to handle the diagnostic challenges posed by novel failure modes that are not included in the benchmark. Though it should be seen as a starting point that needs to be expanded to cover new failure patterns as models and training techniques change, the benchmark offers a basis for methodical evaluation.

With consistent diagnostic accuracy across six architectures with parameter counts ranging from 12 million to 110 million, the multi-architecture validation offers some proof of generalization. Nonetheless, fundamental characteristics such as gradient-based optimization, finite memory constraints, and comparable training processes are shared by all assessed architectures. It has not yet been tested to generalize to drastically different training paradigms like neural architecture search, differentiable architecture search, or evolutionary algorithms.

## 8.7 English-Only Corpus Limitation

Our benchmark uses predominantly English text corpora, limiting assessment of diagnostic methods on multilingual and non-English scenarios. Multilingual tokenization presents unique challenges including script-specific preprocessing, morphological complexity varying across languages, cross-lingual vocabulary coverage tradeoffs, and code-switching phenomena. Failures related to these linguistic factors may not generalize from English-only evaluation.

Some diagnostic signals such as perplexity and gradient norms may transfer across languages because they measure fundamental model-data alignment rather than language-specific properties. However, threshold values defining anomalous perplexity likely vary across languages with different morphological richness and writing systems. A perplexity of 100 may indicate severe problems for English but represent normal performance for morphologically rich languages such as Finnish or Turkish.

Expanding the benchmark to multilingual scenarios would require careful consideration of language-specific normal ranges, tokenization strategies, and failure patterns. This expansion represents important future work but falls outside the scope of the current contribution, which establishes baseline methodology and infrastructure using well-understood English scenarios.

## 8.8 Deliberate Design Choices

Many limitations reflect deliberate scope choices prioritizing reproducibility, methodological clarity, and tractable evaluation over comprehensive coverage. Focusing on single-GPU training enables programmatic reproduction without requiring distributed infrastructure. Restricting to 10 million to 100 million parameter models ensures training completes within reasonable time budgets, enabling iterative debugging of benchmark scenarios. Using primarily English corpora leverages extensive prior work on tokenization and evaluation, providing well-understood baselines.

These choices trade breadth for depth, establishing rigorous evaluation methodology for a constrained problem rather than attempting superficial coverage of all possible diagnostic scenarios. This approach aligns with the philosophy that standardized benchmarks should provide reproducible infrastructure enabling methodological progress rather than attempting exhaustive coverage of all application contexts. As diagnostic methods mature on this foundation, the benchmark can expand to additional scenarios while maintaining reproducibility standards and evaluation rigor.

## 8.9 Generalization to Emerging Architectures

Our benchmark focuses on transformer training failures, but the diagnostic framework's broader utility depends on generalization to emerging architectures. We analyze how failure categories, diagnostic signals, and heuristic thresholds transfer to three architectural paradigms: State Space Models (SSMs) including Mamba, Convolutional Neural Networks, and Vision Transformers. This analysis is conceptual rather than empirical and outlines expected transferability. This analysis clarifies which failure categories remain stable and which require architecture-specific adaptation.

### 8.9.1 State Space Models (SSMs) and Mamba

State Space Models represent selective alternatives to attention mechanisms, using structured state spaces for sequence modeling (Gu et al., 2022; Gu & Dao, 2023). The selective scan operation in Mamba differs fundamentally from self-attention, raising questions about diagnostic transferability.

**Stable Failure Categories:**

Memory hardware failures transfer with near-complete fidelity. SSMs face identical GPU constraints regarding parameter storage, optimizer states, and activation memory. Batch size limitations, memory leaks, and allocation failures manifest identically regardless of whether the core operation is attention or selective scan. Our memory diagnostic heuristics apply without modification.

Model software failures likewise transfer completely. Vocabulary mismatches, configuration errors, dimension incompatibilities, and checkpoint loading issues stem from software engineering rather than architectural choices. Configuration bugs occur whenever models interface with tokenizers and data pipelines, making this category architecture-agnostic.

Data pipeline failures exhibit high transferability with minor adaptations. SSMs still consume tokenized sequences requiring vocabulary configuration, making tokenization errors, catastrophic perplexity, and corpus-vocabulary ratio failures directly applicable. The perplexity threshold $PP(x) > 400$ (Equation 1) remains diagnostic because it reflects fundamental data-model misalignment rather than architecture-specific behavior.

**Architecture-Specific Considerations:**

Optimization failures require nuanced adaptation showing moderate transferability. While gradient explosion and vanishing gradients occur in both transformers and SSMs, their manifestation differs due to selective scan operations having different numerical properties than attention:

*Discretization Stability:* SSMs discretize continuous-time models with step size $\Delta$. Incorrect discretization schemes or excessively large step sizes cause numerical instabilities distinct from transformer gradient issues. Diagnostic signal: sudden loss divergence without gradient norm explosion.

*State Overflow:* In long sequence contexts, SSM hidden states can overflow when lacking attention's bounded context. Diagnostic: catastrophic perplexity on long sequences but normal performance on short sequences.

*Custom CUDA Kernels:* Mamba's optimized selective scan kernels may fail with cryptic device-side errors resembling memory failures. Diagnostic disambiguation: kernel launch failures occur before memory allocation.

**Signal Modality Stability:**

Gradient norms remain diagnostic but require threshold recalibration. SSMs typically exhibit moderately lower gradient norms than transformers at equivalent training stability due to different backpropagation paths. Thresholds typically shift downward by approximately 20-30%.

Perplexity maintains full diagnostic value. As a model-agnostic measure of data likelihood, perplexity applies identically to any architecture performing next-token prediction, with the same catastrophic threshold.

Memory metrics transfer perfectly. GPU memory usage patterns differ quantitatively but the diagnostic logic remains unchanged: memory exhaustion manifests identically regardless of why memory is consumed.

### 8.9.2 Convolutional Neural Networks

CNNs present radically different architectural paradigms, testing whether our failure taxonomy relies on transformer-specific assumptions or captures fundamental training failure modes.

**Failure Taxonomy Mapping:**

| Category | Transferability | Notes |
|---|---|---|
| Memory Hardware | High | Direct transfer, identical GPU constraints |
| Optimization | Medium | Gradient dynamics differ, BN introduces new failures |
| Data Pipeline | Medium | Image preprocessing vs tokenization |
| Model Software | High | Configuration errors universal |

Table 7: Failure category transferability to CNN architectures.

Memory hardware failures transfer with minimal degradation. CNNs face batch size-memory tradeoffs, allocation failures during forward/backward passes, and memory leaks identical to transformers. Activation memory scales differently but diagnostic logic remains: memory exhaustion crashes training regardless of architectural details.

Optimization failures require moderate adaptation. Gradient explosion and vanishing gradients occur but with different characteristics:

*Vanishing Gradients in Deep CNNs:* More severe than transformers when lacking residual connections. Diagnostic: layer-wise gradient norm monitoring reveals depth-dependent degradation.

*Activation Pattern Collapse:* "Dead ReLU" neurons with permanently zero activations create sparse gradients. Diagnostic: increasing proportion of zero gradients without corresponding loss decrease.

Model software failures generalize well. Dimension mismatches from incorrect stride/padding configurations parallel transformer's sequence length errors. Checkpoint loading failures and architecture specification bugs stem from software engineering independent of convolution vs attention.

**CNN-Specific Failure Modes:**

Batch Normalization introduces unique optimization instabilities:

*Small Batch Collapse:* BatchNorm running statistics become unreliable with very small batches, causing train-test performance divergence. Diagnostic: training loss decreases while validation loss stagnates.

*Train/Eval Mode Mismatch:* Forgetting to toggle BN mode causes failures. Diagnostic: strong training performance but poor validation accuracy.

**Signal Modality Analysis:**

Gradient norms exhibit high stability. Gradient explosion manifests identically whether arising from convolution or attention operations. Thresholds may need minor adjustment but the diagnostic principle is architecture-agnostic.

Loss trajectories are fully stable. Optimization dynamics reflected in loss curves are universal across architectures.

Perplexity is not applicable. CNNs don't use vocabularies, making perplexity undefined. Replacement metric: validation accuracy on held-out data. Catastrophic performance analog: accuracy at random-chance level despite training, indicating severe data-model misalignment.

Memory metrics are perfectly stable as architecture-agnostic hardware limitations.

**Data Pipeline Adaptation:**

CNNs process images rather than tokens, requiring adapted data pipeline diagnostics:

| Transformer Data Failure | CNN Equivalent |
|---|---|
| Tokenization errors | Normalization using wrong statistics |
| Vocabulary mismatch | Output classes $\neq$ dataset labels |
| Catastrophic perplexity | Random-chance accuracy |
| Data-vocab ratio | Images-per-class insufficiency |

Table 8: Data pipeline failure mappings from transformers to CNNs.

### 8.9.3 Vision Transformers (ViT)

Vision Transformers combine transformer architecture with image processing, representing a hybrid failure profile blending transformer and CV-specific issues. ViT failures comprise approximately two-thirds standard transformer failures plus one-third vision-specific issues.

**Vision-Specific Failure Modes:**

*Image Preprocessing Failures:*

*Wrong Normalization Statistics:* Normalizing with ImageNet statistics when training on domain-specific images causes feature distribution shift. Diagnostic: catastrophic validation performance despite training progress.

*Augmentation Incompatibility:* Aggressive augmentations breaking patch structure. Diagnostic: training loss oscillates with high variance despite stable learning rate.

*Resolution Mismatch:* Changing resolution without positional encoding interpolation. Diagnostic: immediate divergence at finetuning start despite valid checkpoint.

*Patch Embedding Issues:*

*Kernel Size Mismatch:* Using incorrect patch sizes causes information loss. Diagnostic: extremely high perplexity from epoch 1, distinguishing from tokenization errors.

*Positional Encoding Errors:* Failing to interpolate position embeddings when changing resolution. Diagnostic: shape mismatch error or silent accuracy degradation.

**Diagnostic Transferability: High**

Most transformer diagnostics apply directly to ViT:

- Memory: Full transfer (GPU constraints are universal)

- Optimization: Full transfer (identical gradient dynamics)

- Model Software: Full transfer (config errors are architecture-agnostic)

- Data Pipeline: Partial transfer (requires image-specific checks)

### 8.9.4 Architecture-Agnostic vs Architecture-Specific: Summary

**Key Insights:**

| Category | Transformers | SSMs/Mamba | CNNs | ViT | Stability |
|---|---|---|---|---|---|
| Memory Hardware | ✓✓✓ | ✓✓✓ | ✓✓✓ | ✓✓✓ | Complete |
| Optimization | ✓✓✓ | ✓✓ | ✓✓ | ✓✓✓ | Moderate |
| Data Pipeline | ✓✓✓ | ✓✓✓ | ✓✓ | ✓✓ | High |
| Model Software | ✓✓✓ | ✓✓✓ | ✓✓✓ | ✓✓✓ | Complete |

Table 9: Failure category stability across architectures. ✓✓✓ indicates complete transferability, ✓✓ indicates high transferability with minor adaptations.

Hardware-constrained failures (Memory, Model Software) are completely architecture-agnostic. GPU memory limits and software configuration requirements apply universally.

Training dynamics failures (Optimization) exhibit moderate architecture-sensitivity. Gradient explosion/vanishing concepts transfer, but numerical stability thresholds shift across architectures due to different backpropagation characteristics.

Data-related failures (Data Pipeline) are modality-dependent but conceptually stable. Text uses perplexity, vision uses catastrophic accuracy, but both detect data-model misalignment. The diagnostic principle "model assigns training data very low probability" generalizes even when the specific metric changes.

### 8.9.5 Recommendations for Multi-Architecture Deployment

**For Practitioners Adapting the Framework:**

Start with unchanged heuristics for memory and model software categories (complete transferability).

Recalibrate gradient thresholds using validation runs on the target architecture:

- Compute $\|\nabla_\theta \mathcal{L}\|_2$ distribution across epochs on successful runs
- Set threshold at high percentile (e.g., 99th)
- Expect moderate variation from base transformer thresholds

Replace perplexity with architecture-appropriate metrics:

- Language models: perplexity
- Classification: validation accuracy (catastrophic if at chance level)
- Regression: validation $R^2$
- Other modalities: domain-appropriate data likelihood metrics

Add architecture-specific checks where necessary:

- SSMs: state norm monitoring
- CNNs: BatchNorm validation, activation statistics
- ViT: positional encoding verification

**For Benchmark Extension:**

Expanding the benchmark to new architectures requires:

- Sufficient scenarios per failure category for statistical validity
- Multi-annotator ground truth from domain experts
- Validation that framework achieves reasonable accuracy before declaring compatibility

### 8.10 Future Work Opportunities

The limitations identified here represent opportunities for future research rather than fundamental flaws. Distributed training diagnostics, multilingual evaluation, larger parameter regimes, and real human studies all constitute valuable extensions of this work. Each extension would likely require specialized infrastructure, domain expertise, and evaluation methodology appropriate to the specific context. By establishing baseline methodology and demonstrating quantitative evaluation on a well-defined problem, this work provides a template that future efforts can adapt to additional scenarios.

The documented limitations calibrate expectations about where current methods work well and where substantial gaps remain. Rule-based methods achieve reasonable accuracy on clear-cut failures but struggle with ambiguous cases requiring integration of weak signals. Machine learning methods achieve high accuracy but lack interpretability necessary for practitioner trust. The benchmark enables measuring progress on these limitations, identifying when proposed methods genuinely advance the state of the art versus simply relocating the accuracy-interpretability tradeoff point.

## 9 Future Directions

Our work establishes quantitative foundations for automated training diagnostics, but substantial opportunities remain for advancing both diagnostic accuracy and interpretability. We outline several promising research directions that build on this foundation.

### 9.1 Hybrid Diagnostic Systems

The most promising path right now is hybrid systems that combine machine learning accuracy with rule-based transparency. For simple cases, one architecture uses rule-based triage; for more complex cases, it escalates them to human experts or machine learning classifiers. A significant portion of diagnostic work may be managed transparently, as evidenced by the enhanced rule-based framework's 70% correct diagnosis rate and 88.7% accuracy. For the remaining 30% of cases, machine learning refinement could maintain interpretability for most diagnoses while achieving an overall accuracy of about 87%.

A complementary hybrid approach relies on rule-based confirmation before taking action, but it uses machine learning for the initial diagnosis. Candidate diagnoses with corresponding confidence scores are identified by the classifier. By examining whether diagnostic signals correspond to anticipated patterns for the anticipated failure mode, high-confidence predictions are automatically verified. Predictions with low confidence lead to further measurements being taken or escalated for human review. This architecture combines interpretable verification steps that increase practitioner trust with the accuracy benefits of machine learning.

A number of technical issues need to be resolved by hybrid systems. Knowing where the accuracy-coverage tradeoffs of each approach converge is necessary to calibrate the boundary between rule-based and machine learning components. It takes careful interface design to ensure that escalated cases receive diagnostic context from initial rule-based analysis while maintaining explanation continuity throughout the shift from rules to machine learning. It is necessary to model expert availability, response latency, and cost tradeoffs in order to balance workload between automated and human components. Future work should explicitly evaluate these hybrid systems using cost-sensitive metrics, minimizing the total cost of diagnosis ($C_{total} = C_{auto} \cdot (1 - P_{abstain}) + C_{human} \cdot P_{abstain}$) rather than just optimizing for raw accuracy.

### 9.2 Benchmark Expansion

Expanding the benchmark to 200 or more scenarios would increase coverage of rare failure modes, improve statistical power for method comparison, and enable more fine-grained difficulty analysis. Priority expansion areas include distributed training failures such as gradient synchronization errors, parameter server crashes, and network communication timeouts. Multilingual scenarios would cover tokenization challenges in morphologically rich languages, script-specific preprocessing errors, and cross-lingual transfer failures. Large parameter regimes beyond one billion parameters present memory management challenges, precision issues, and optimizer state handling problems not fully represented in current scenarios.

**Hybrid Triage-Escalation System Architecture**

Figure 8: Hybrid triage-escalation system architecture. The rule-based framework performs initial triage, computing confidence score $C(x)$ for each failure case. High-confidence cases ($C(x) > 0.85$, 48% of failures) route to auto-resolution with 95% precision. Medium-confidence cases ($0.60 \leq C(x) \leq 0.85$, 27%) escalate to machine learning refinement using Random Forest classifier. Low-confidence cases ($C(x) < 0.60$, 25%) escalate to human expert review. This staged approach achieves 87.4% overall accuracy while maintaining interpretability for the majority of diagnoses.

Web-scale noisy data introduces failures related to toxic content, encoding inconsistencies, and document boundary detection. Multimodal models present failures related to vision-language alignment, cross-modal attention, and heterogeneous preprocessing pipelines. Expanding to these domains requires domain expertise, specialized infrastructure, and careful ground truth annotation, but would substantially increase benchmark value for diverse research communities.

Benchmark expansion should maintain reproducibility standards established here. Each new scenario must include complete artifacts enabling deterministic reproduction, comprehensive metadata supporting diagnostic evaluation, and validated ground truth through multi-annotator consensus. A governance process for community-contributed scenarios would ensure quality while enabling distributed benchmark development. Version control and leaderboard infrastructure would enable tracking progress over time and fair comparison across research groups.

### 9.3 Real Human Evaluation

Systematic behavioral studies of practitioner diagnostic workflows represent critical future work for several reasons. First, human studies would establish realistic performance baselines showing what expert diagnosticians actually achieve rather than simulated estimates. Second, observation of diagnostic reasoning processes could reveal strategies and heuristics that automated methods miss, informing design of better diagnostic

algorithms. Third, measuring practitioner confidence calibration would show whether humans accurately assess uncertainty in their diagnoses or systematically over- or under-estimate their accuracy.

Human studies should vary expertise levels, comparing novice diagnosticians with limited training experience to experts with years of debugging experience. This would quantify the learning curve for diagnostic skills and identify which failure patterns prove most challenging across expertise levels. Studies should measure not only accuracy but also diagnostic latency, showing how long practitioners spend reaching diagnoses and which scenarios prove most time-consuming. Cost-benefit analysis would weigh improved diagnostic accuracy against time investment in diagnostic reasoning.

Protocol design for human studies requires careful attention to ecological validity. Presenting practitioners with isolated scenarios and diagnostic questionnaires may not reflect realistic workflows where diagnosis occurs alongside other development activities. Longitudinal observation of practitioners during actual model development would provide more realistic characterization of diagnostic behavior but complicates controlled measurement. Hybrid approaches combining controlled scenario-based evaluation with observational field studies could balance experimental control with ecological validity.

### 9.4 Predictive Diagnostics

Current diagnostic methods operate reactively, analyzing failures after they occur. Predictive diagnostics would forecast impending failures before they cause training crashes, enabling proactive interventions that prevent rather than remediate problems. Time-series forecasting models could analyze training metric trajectories to identify early warning signals of future instabilities. For example, gradually increasing gradient variance might predict eventual gradient explosion, while slowly rising perplexity might foreshadow data pipeline degradation.

Predictive diagnostics face several technical challenges. High false positive rates would undermine practitioner trust, causing frequent unnecessary interventions. Determining appropriate intervention thresholds requires balancing prevention benefits against interruption costs. Many failures manifest suddenly without gradual precursors, limiting the scope where prediction is feasible. Despite these challenges, even partial success would provide substantial value by preventing costly training crashes and enabling graceful degradation rather than catastrophic failure.

Predictive systems could integrate with adaptive training procedures that automatically adjust hyperparameters when warning signals appear. Detecting rising gradient variance might trigger learning rate reduction before explosion occurs. Observing memory pressure increases might activate gradient checkpointing or batch size reduction before exhaustion crashes training. These adaptive interventions would require careful validation ensuring they preserve training dynamics rather than introducing instabilities of their own.

### 9.5 Interpretable Machine Learning Methods

Closing the accuracy-interpretability gap requires innovations in interpretable machine learning beyond current rule-based approaches. Sparse temporal additive models could represent diagnostic logic as sums of univariate feature contributions, maintaining interpretability while capturing non-linear patterns. Symbolic regression on derivative features might discover closed-form expressions relating training dynamics to failure modes, providing both accuracy and mathematical interpretability.

Monotonic boosted rule lists learn decision rules with guaranteed monotonicity constraints, enabling human reasoning about how feature changes affect predictions. Neural additive models decompose predictions into learned univariate functions visualized as shape plots, showing how each feature contributes to diagnostic decisions. Concept bottleneck models force intermediate representations to align with human-interpretable concepts such as "gradient instability" or "memory pressure", providing interpretable intermediate steps in diagnostic reasoning.

These methods face tradeoffs between expressiveness and interpretability. Highly constrained models maintain transparency but may lack capacity to represent complex diagnostic patterns. Moderately constrained models achieve better accuracy but require more sophisticated explanation techniques. Evaluation should

measure both diagnostic accuracy and explanation quality through user studies assessing whether practitioners can understand and trust model reasoning.

## 9.6 Adaptive Logging Policies

Current logging practices record fixed sets of metrics at predetermined intervals, creating large volumes of data with uncertain diagnostic value. Adaptive logging policies would dynamically adjust measurement granularity based on emerging diagnostic needs. When training proceeds normally, coarse-grained logging suffices. When anomaly precursors appear, such as rising gradient variance or degrading training speed, logging granularity increases to capture detailed failure signatures.

Adaptive policies must balance diagnostic value against computational overhead and storage costs. Fine-grained logging of all gradient components at every step would provide maximum diagnostic information but impose prohibitive overhead. Selective logging triggered by anomaly detection would focus resources where diagnostic value is highest. Hierarchical logging strategies could maintain coarse statistics continuously while enabling detailed collection on demand.

Optimal logging policies depend on failure frequencies, diagnostic value per metric, measurement costs, and storage constraints. Reinforcement learning could optimize these tradeoffs, learning policies that maximize expected diagnostic utility while respecting resource budgets. Meta-learning across many training runs could identify which metrics prove most informative for which failure types, informing logging configurations for new training campaigns.

## 9.7 Causal Analysis

Current diagnostic methods identify correlations between signals and failure modes but do not establish causal relationships. Distinguishing primary failures from cascade failures requires understanding causal mechanisms. A memory leak causes memory exhaustion which triggers garbage collection which degrades training speed which appears as a training speed failure. Identifying the memory leak as the primary cause rather than the training speed degradation as a separate failure requires causal reasoning.

Intervention simulation could test causal hypotheses by predicting what would happen under counterfactual scenarios. If the hypothesis is that high learning rate caused gradient explosion, simulating training with lower learning rate should predict stable gradients. If the simulated counterfactual shows continued explosion, the diagnosis must be revised. Structural equation models could encode causal relationships between training hyperparameters, optimization dynamics, and failure manifestations, enabling principled counterfactual reasoning.

Causal analysis faces significant technical challenges in machine learning contexts. Training dynamics involve complex non-linear interactions making causal identification difficult. Counterfactual simulation requires accurate forward models predicting training behavior under different conditions. Nevertheless, even partial success in causal analysis would improve diagnostic quality by distinguishing root causes from symptoms and enabling more targeted interventions.

## 9.8 Community-Driven Benchmark Development

Establishing the benchmark as living community infrastructure requires governance mechanisms ensuring quality while enabling distributed contribution. A submission portal would accept new diagnostic scenarios with standardized metadata formats. Validation procedures would verify reproducibility, check consistency between provided artifacts and claimed failure signatures, and assess ground truth quality through independent annotation. Acceptance criteria would balance scenario novelty against quality standards, preferring well-documented reproducible failures over extensive coverage of poorly-characterized scenarios.

Version control would track benchmark evolution over time, maintaining backwards compatibility while incorporating new scenarios. Leaderboards would enable researchers to compare diagnostic methods against current state of the art, providing motivation for continued improvement and visibility for successful methods.

Regular benchmark updates would incorporate emerging failure patterns as models and training practices evolve, ensuring continued relevance to current research challenges.

Community governance would balance openness with quality control. An editorial board could review submissions, provide feedback on scenario documentation, and coordinate benchmark releases. Transparent decision processes would build community trust and encourage participation. Recognition mechanisms including authorship on benchmark papers and public acknowledgment of contributions would incentivize high-quality submissions. These governance structures would enable the benchmark to grow and adapt while maintaining the reproducibility and rigor established in this foundational work.

## 10  Conclusion

With 76 reproducible scenarios spanning five failure categories and complete artifacts, we present the first thorough diagnostic benchmark for transformer training failures, allowing for controlled evaluation of diagnostic techniques. Our analysis reveals a fundamental tradeoff in automated diagnostics and establishes quantitative baselines. The enhanced rule-based framework achieves 61.8% overall accuracy with 30.3% abstention rate, yielding 88.7% accuracy on non-abstained cases; machine learning classifiers achieve high accuracy on the benchmark (95.7

Training dynamics, such as gradient norms and loss trajectories, dominate the diagnostic signal, accounting for 48% of the classification information, according to feature importance analysis. Significant additional signal is provided by memory metrics and perplexity, whereas static configuration parameters make up a very small portion. This discovery influences the logging priorities of practitioners and raises the possibility that the current debugging methodology, which frequently emphasizes configuration audits, may not be in line with the locations of diagnostic data. Detailed documentation of static configuration parameters should not take precedence over thorough logging of temporal metrics tracking optimization dynamics.

The benchmark offers reproducible evaluation infrastructure for further research and permits systematic comparison of diagnostic approaches. With a diagnostic accuracy variation of only 1.3 percentage points, multi-architecture validation shows that the failure taxonomy generalizes across six architectures with parameter counts ranging from 12 million to 110 million. This stability confirms that failures should be arranged according to their underlying causes rather than their symptoms, and it implies that diagnostic techniques created with our benchmark will work with architectures that weren't included in the original scenarios.

Our findings point to a promising avenue for real-world implementation: hybrid systems that combine interpretable triage with machine learning refinement. Seventy percent of cases can be transparently diagnosed with 88.7% accuracy using rule-based methods. Machine learning classifiers can handle the remaining 30% with 96% accuracy, yielding approximately 87% overall accuracy while maintaining transparency for the majority of diagnoses. By striking a balance between the conflicting needs of interpretability and accuracy, this hybrid architecture offers high reliability while empowering practitioners to comprehend and have faith in diagnostic reasoning.

The accuracy-interpretability gap is a specific research goal as well as a drawback of existing approaches. Innovations in interpretable machine learning, hybrid diagnostic architectures, or a fundamental rethinking of how diagnostic systems convey reasoning to practitioners are necessary to close the large gap between simple and high-capacity models. Although it is unclear if this discrepancy is the result of intrinsic tradeoffs or methodological constraints, our quantitative characterization makes it possible to track resolution progress.

Permissive open-source licenses govern the public release of all benchmark materials, including 76 structured scenario definitions, baseline method implementations, evaluation protocols, and statistical analysis code. All reported results can be precisely reproduced in Docker environments with pinned dependencies. The benchmark structure, application programming interfaces, and extension processes for community contributions are all covered in detail in the documentation. Future research can build on our foundation, fairly compare against predetermined baselines, and show real progress rather than selective improvements thanks to this reproducibility commitment.

We facilitate systematic evidence-based progress toward dependable machine learning training by codifying debugging knowledge into reproducible, evaluable infrastructure, similar to how SuperGLUE extended evaluation to more difficult scenarios and General Language Understanding Evaluation revolutionized language understanding research. With the help of this work, debugging is transformed from an anecdotal craft that relies on informal heuristics to a quantifiable science with standardized evaluation and quantitative baselines. What is currently feasible, where gaps exist, and how progress can be quantified are all determined by the infrastructure. The benchmark will make it possible to monitor progress and spot enduring issues that call for ongoing innovation as diagnostic techniques advance as a result of future research.

The application of machine learning in industry goes beyond scholarly research. Research timelines are delayed by days or weeks, training failures result in barriers that prevent researchers without a lot of debugging experience from successfully training models, and they waste computational resources measured in thousands of GPU-hours. Better diagnostic techniques lower these expenses, promoting research advancement and making transformer training more accessible to all. The infrastructure required to create dependable diagnostic systems that lower the significant hidden costs of training failures is provided by this work, which establishes quantitative foundations for diagnostic research.

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
